**EMBO**
**Molecular Medicine**

# MicroRNA-574 regulates FAM210A expression and influences pathological cardiac remodeling

Jiangbin Wu[1,†] ID, Kadiam C Venkata Subbaiah[1,†], Feng Jiang[1,2] ID, Omar Hedaya[1,2], Amy Mohan[1], Tingting Yang[3], Kevin Welle[4], Sina Ghaemmaghami[4], Wai Hong Wilson Tang[5], Eric Small[1], Chen Yan[1] & Peng Yao[1,2,6,7,*]

## Abstract

Aberrant expression of mitochondrial proteins impairs cardiac function and causes heart disease. The mechanism of regulation of mitochondria encoded protein expression during cardiac disease, however, remains underexplored. Here, we show that multiple pathogenic cardiac stressors induce the expression of miR-574 guide and passenger strands (miR-574-5p/3p) in both humans and mice. miR-574 knockout mice exhibit severe cardiac disorder under different pathogenic cardiac stresses while miR-574-5p/3p mimics that are delivered systematically using nanoparticles reduce cardiac pathogenesis under disease insults. Transcriptomic analysis of miR-574-null hearts uncovers family with sequence similarity 210 member A (FAM210A) as a common target mRNA of miR-574-5p and miR-574-3p. The interactome capture analysis suggests that FAM210A interacts with mitochondrial translation elongation factor EF-Tu. Manipulating miR-574-5p/3p or FAM210A expression changes the protein expression of mitochondrial-encoded electron transport chain (ETC) genes but not nuclear-encoded mitochondrial ETC genes in both human AC16 cardiomyocyte cells and miR-574-null murine hearts. Together, we discovered that miR-574 regulates FAM210A expression and modulates mitochondrial-encoded protein expression, which may influence cardiac remodeling in heart failure.

**Keywords** cardiac remodeling; FAM210A; gene regulation; microRNA; mitochondria

**Subject Category** Cardiovascular System

## Introduction

Heart failure (HF) is a leading cause of morbidity and mortality worldwide (Ambrosy *et al*, 2014). The current 5-year mortality following diagnosis with HF is 52.6% (Benjamin *et al*, 2018). An important cause of HF is pathological cardiac remodeling, which can be triggered by hypertension, coronary artery disease, chemotherapy, myocardial infarction (MI), valvular disease, or rare and damaging genetic mutations. Following infarction and pressure overload, hypertrophic growth of cardiomyocytes (CMs), along with CM apoptosis and cardiac fibroblast (CF) proliferation and cardiac fibrosis, pathologically remodels the size, mass, geometry, and function of a heart, leading to HF. This pathological remodeling process is accompanied by changes in the expression of mitochondrial genes and specific miRNAs (Abdellatif, 2012; van Rooij & Olson, 2012; Lu & Wang, 2018; Zhou & Tian, 2018).

In mammals, the heart is the most mitochondria-rich organ. Mitochondria play a critical role in metabolism, cell proliferation, apoptosis, and CM contractility. The electron transport chain (ETC) complex, located in the mitochondrial inner membrane, generates an electrochemical-proton gradient that drives the synthesis of ATP. The synthesis of mitochondrial ETC complex proteins is tightly regulated in order to maintain mitochondrial homeostasis and normal cardiac function (Couvillion *et al*, 2016). Accordingly, aberrant synthesis of mitochondrial proteins impairs heart function and causes heart disease. Early-stage cardiac hypertrophy induces mitochondrial protein translation and activity as a compensatory response, which may promote pathological cardiac remodeling (Zhou *et al*, 2013). In contrast, endpoint HF is accompanied by reduced mitochondrial protein synthesis and mitochondrial dysfunction (Brown *et al*, 2017). Thus, the restoration of mitochondrial function by targeting biogenesis pathways of mitochondrial proteins is thought to be an important strategy to develop therapeutics for

1   Department of Medicine, Aab Cardiovascular Research Institute, University of Rochester School of Medicine & Dentistry, Rochester, New York, NY, USA
2   Department of Biochemistry & Biophysics, University of Rochester School of Medicine & Dentistry, Rochester, New York, NY, USA
3   Department of Ophthalmology, Columbia University, New York, NY, USA
4   Mass Spectrometry Resource Lab, University of Rochester School of Medicine & Dentistry, Rochester, New York, NY, USA
5   Department of Cardiovascular Medicine, Cleveland Clinic, Cleveland, OH, USA
6   The Center for RNA Biology, University of Rochester School of Medicine & Dentistry, Rochester, New York, NY, USA
7   The Center for Biomedical Informatics, University of Rochester School of Medicine & Dentistry, Rochester, New York, NY, USA
    *Corresponding author. Tel: +1 585 276 7708; E-mail: peng_yao@urmc.rochester.edu
    †These authors contributed equally to this work

heart disease (Kwong & Molkentin, 2015; Ping *et al*, 2015; Wang *et al*, 2016a; Brown *et al*, 2017). This approach, however, requires a better understanding of the regulatory mechanisms of mitochondrial genes. ETC complex genes contain both mitochondrial-encoded (MEGs) and nuclear-encoded mitochondrial genes (NEMGs). Many studies have examined mechanisms underlying transcriptional regulation of these genes (Wiesner *et al*, 1994; Wu *et al*, 1999; Goffart *et al*, 2004; Gleyzer *et al*, 2005; Cotney *et al*, 2007; Topf *et al*, 2016; Eisenberg-Bord & Schuldiner, 2017; Cardamone *et al*, 2018). The role of regulation of protein expression of NEMGs and MEGs in cardiac disease has received little attention until recently. It has been shown that NEMG protein expression was enhanced in the murine heart with pressure overload (Zhou *et al*, 2013). Despite this finding, the regulation of MEG protein expression and the consequent coupling with NEMG protein expression in the heart remains largely underexplored.

miRNAs are 18–25-nt small non-coding RNAs that regulate gene expression via mRNA decay and/or translational repression. miRNAs associate with Argonaute (Ago) proteins to form the RNA-induced silencing complex (RISC) and guide miRISCs to specific mRNA targets (Bartel, 2009). Global loss of miRNAs in the murine heart via heart-specific deletion of Dicer produced a poorly developed ventricular myocardium, highlighting the importance of miRNAs in the heart (Chen *et al*, 2008). Furthermore, abnormal expression of miRNAs has been observed frequently in response to pathological stress that impedes CM function and causes cardiac hypertrophy and HF (van Rooij *et al*, 2006; van Rooij *et al*, 2008; Lu & Wang, 2018). Some miRNAs were explored as therapeutic targets to prevent HF (Thum *et al*, 2008; Montgomery *et al*, 2011; Hullinger *et al*, 2012; Boon *et al*, 2013), while other miRNAs have been found to protect the heart from cardiac pathogenesis (van Rooij *et al*, 2008; Aurora *et al*, 2012; Eulalio *et al*, 2012). Inspired by these findings, stable miRNA mimics and antagonists for miRNAs have been developed to prevent or reverse various heart diseases in experimental HF mouse models (van Rooij & Olson, 2012) and human clinical trials (e.g., miR-29 mimics and miR-92 inhibitors, miRagen Therapeutics, inc.). Previous studies have shown that miRNAs can regulate the translation of MEG mRNAs to affect heart function. Mitochondrial-localized miR-1 (Zhang *et al*, 2014b) and miR-21 (Li *et al*, 2016), for instance, directly target a selective cohort of seed sequence-specific MEG mRNAs and, respectively, activate their translation in skeletal muscle cells and cardiac myocytes. However, whether miRNAs also act through the regulation of general MEG protein expression to influence heart function is unclear.

In this study, we have discovered that miR-574 regulates FAM210A (family with sequence similarity 210 member A) expression and antagonizes pathological cardiac remodeling. miR-574 null mice exhibit severe cardiac dysfunction under stress conditions, and exogeneous delivered miR-574 mimics protect against pathogenesis. Mechanistically, FAM210A functions as a novel regulatory factor of mitochondrial-encoded protein expression. Both the guide strand miR-574-5p and the passenger strand miR-574-3p of miR-574 target FAM210A to modulate the expression of mitochondrial-encoded ETC proteins. This novel mechanism may contribute to the maintenance of mitochondrial homeostasis and prevent pathological cardiac remodeling.

# Results

## Cardiac stress induces miR-574-5p and miR-574-3p in human and mouse hearts

From mining unbiased screening data from four laboratories, the guide strand miR-574-5p has been identified as an miRNA that is robustly induced in the heart in response to pathological cardiac remodeling, which included the cardiac tissues of patients during the initial stages following MI onset (Bostjancic *et al*, 2009), a mouse model of left coronary artery occlusion-derived MI (van Rooij *et al*, 2006), and aged murine hearts (Fig 1A) (Boon *et al*, 2013). On the other hand, the passenger strand miR-574-3p is increased in the human hearts after MI and more remarkably induced by exercise training in murine hearts (Liu *et al*, 2015). However, the cardiac functions of complementary strands of miR-574-5p and miR-574-3p remain unknown. miR-574 is conserved in 43 animal species (10 primates and 33 mammals, UCSC Genome Browser). In mammals, miR-574 is located in intron 1 of the host gene FAM114A1 (Family with sequence similarity 114 member A1; Fig 1B). We confirmed that both miR-574-5p and miR-574-3p were significantly induced in heart tissues of chronic HF patients, with the expression level of miR-574-5p even higher than that of miR-574-3p (Fig 1C). Northern blot analysis showed that both miR-574-5p and miR-574-3p were expressed in the normal murine heart (Fig EV1A). miR-574-5p and miR-574-3p were both significantly induced in isoproterenol (ISO; 4 weeks)- and transverse aortic constriction (TAC; 4 weeks)-treated mouse hearts, compared to vehicle treatment and sham operation, respectively (Fig 1D and E). miR-574-5p was more dominantly expressed in isolated primary mouse adult cardiomyocytes (ACMs) in comparison to miR-574-3p (Fig EV1B). miR-574-5p and miR-574-3p were significantly induced in ACMs and adult cardiac fibroblasts (ACFs), respectively, while the other strand of miRNA was only slightly induced in each cell type from mice with TAC surgery (Fig EV1C). We detected ~2- to 3-fold increase of miR-574-5p and miR-574-3p in murine hearts during the period of 3–21 days post-TAC surgery (Fig EV1D and E). Moreover, in primary mouse ACMs, both strands of miR-574 were robustly induced by ISO treatment, with the guide strand showing higher induction than the passenger strand (Fig EV1F), which is consistent with the results in the TAC model (Fig EV1C).

## miR-574 gene deletion exacerbates pathological cardiac remodeling

To determine whether miR-574 is protective or harmful in the adult murine heart, we generated a miR-574 knockout (KO) mouse using the embryonic stem (ES) cell clone harboring targeted miR-574 deletion (Appendix Fig S1A; Prosser *et al*, 2011). The allele contained two loxP sites flanking the puromycin resistant cassette that replaces miR-574. The puromycin cassette was removed globally by crossing with ZP3-Cre mice. The complete deletion of the genomic region between the two loxP sites was confirmed by PCR of genomic DNA (Appendix Fig S1B). Using RT–qPCR of RNA from adult murine hearts, we confirmed that the expression of miR-574-5p and miR-574-3p was abolished in the homozygous miR-574 KO mice (Appendix Fig S1C) while expression of the host gene *Fam114a1* remained unchanged in miR-574$^{-/-}$ hearts or isolated ACMs

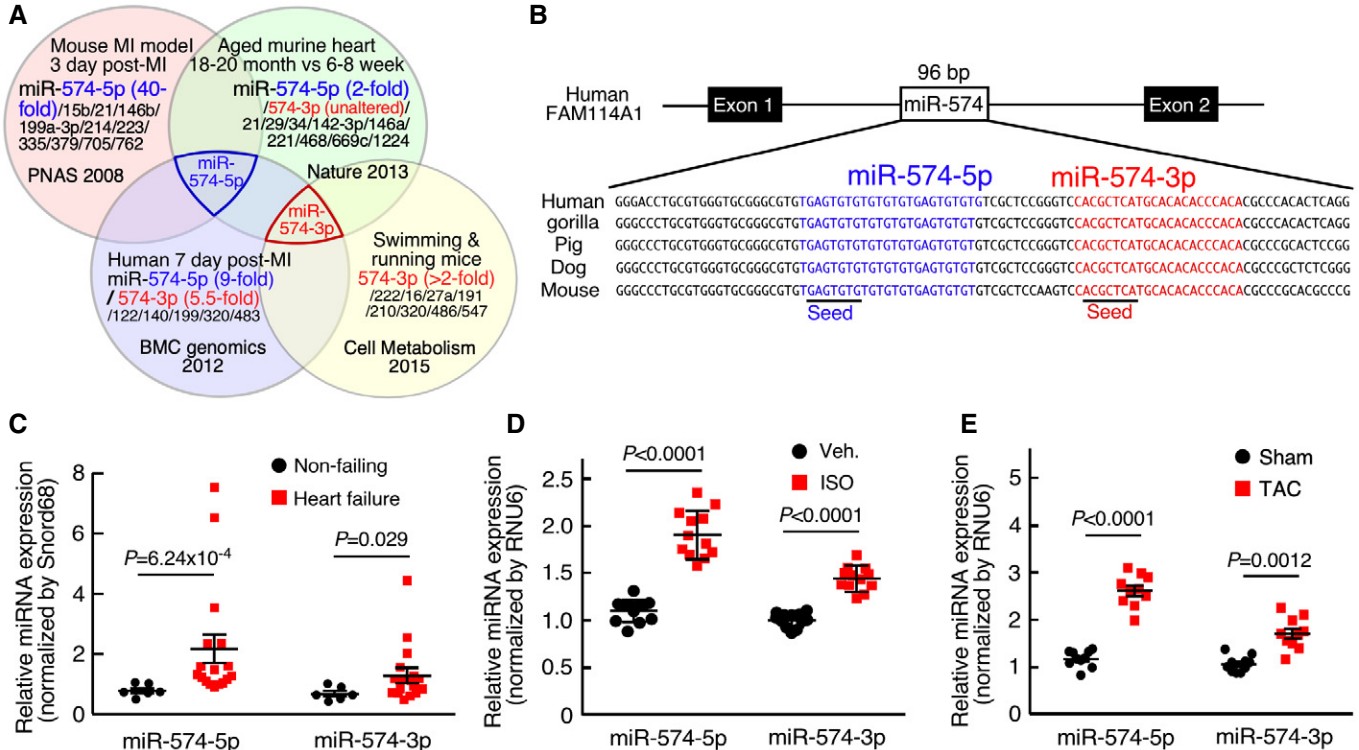

**Figure 1. miR-574-5p and miR-574-3p expression is induced in human and mouse failing hearts.**

A miRNA profiling identified miR-574-5p as the commonly upregulated miRNA in the hearts of aged mice, MI mice, and patients; and miR-574-3p is induced in exercised hearts and MI patients.

B Sequence and location of miR-574 gene in the human genome.

C miR-574-5p and miR-574-3p were highly expressed in the explanted failing hearts from chronic HF patients ($n = 17$) versus donor non-failing hearts ($n = 6$).

D miR-574-5p and miR-574-3p were induced in the hearts of mice with isoproterenol (ISO) infusion (4 weeks; $n = 10$–12).

E miR-574-5p and miR-574-3p were induced in the hearts of mice under transverse aortic constriction (TAC) surgery (4 weeks; $n = 8$–10).

Data information: Data were presented as mean ± SEM. P values were calculated by Mann–Whitney test (C) and unpaired two-tailed Student $t$ test (D, E).
Source data are available online for this figure.

(Appendix Fig S1D). miR-574$^{-/-}$ mice were viable and appeared normal in weight, fertility, and behavior. We conclude that miR-574 is not required for viability and development in mice in the absence of stress.

We then investigated whether miR-574 affects cardiac remodeling in response to chronic β-adrenergic stimulation. Isoproterenol (ISO), a β-adrenergic receptor agonist, was administered to mice via subcutaneous injection (30 mg/Kg/day) for 4 weeks. The hearts of WT and miR-574$^{-/-}$ mice were similar in size at baseline (Fig 2A). However, mutant mice showed a more significant ISO-induced increase in the left ventricle (LV) wall thickness and heart weight/tibia length (HW/TL) ratio, compared to WT mice (Fig 2A and B). miR-574$^{-/-}$ mice exhibited an enhanced cardiac hypertrophy phenotype associated with more enlarged CMs compared to WT mice after ISO treatment (Fig 2C). Picrosirius red staining confirmed that ISO administration resulted in more fibrosis in miR-574$^{-/-}$ than in WT mice (Fig 2D). A hallmark of pathological hypertrophy and HF is the reactivation of a set of fetal cardiac genes, including the hypertrophic gene markers *ANF*, *BNP*, *Myh6*, and *Myh7*, and the fibrosis markers *Col1a2* and *Col3a1*. At baseline level, the

expression of fetal cardiac genes was not altered in miR-574$^{-/-}$ mice (Appendix Fig S1E). However, the expression of these genes was significantly increased in miR-574$^{-/-}$ mice after ISO treatment (Fig 2E), suggesting that miR-574$^{-/-}$ mice are more susceptible to cardiac remodeling than WT mice in response to β-adrenergic stimulation. Meanwhile, we isolated primary ACMs from miR-574$^{-/-}$ and WT mice and treated cells with ISO. miR-574$^{-/-}$ ACMs exhibited more severe ISO-induced hypertrophy (Fig EV2A) and were prone to excessive ISO-induced apoptosis (Fig EV2B) compared to those from WT mice. In addition, miR-574$^{-/-}$ mice showed more severe myocyte apoptosis and more pronounced ROS production than WT mice in the heart (Figs 2F and Fig EV2C).

To confirm the role of miR-574 in pathological cardiac remodeling, we performed transverse aortic constriction (TAC) to model pressure overload-induced cardiac hypertrophy in WT and miR-574$^{-/-}$ adult mice. miR-574$^{-/-}$ mice exhibited more pronounced cardiac hypertrophy 4 weeks after initiation of TAC compared to WT mice, as indicated by HW/TL ratios (Fig 3A and B) and CM cross-sectional area (Fig 3C). Cardiac fibrosis was also exaggerated in miR-574$^{-/-}$ mice subjected to TAC, as demonstrated by

picrosirius red staining (Fig 3D). Consistent with the observed pathological changes, cardiac hypertrophy and fibrosis marker gene expression were increased more significantly in miR-574$^{-/-}$ mice than in WT mice 4 weeks after TAC (Fig 3E). Cardiac function of WT and miR-574$^{-/-}$ mice in response to this cardiac stress was assessed by echocardiography. This demonstrated a significantly impaired fractional shortening (FS) in miR-574$^{-/-}$ mice compared to WT mice (Fig 3F, and Table 1). Also, ROS production and cardiac apoptosis (though occurring at modest level under pressure overload) were more pronounced in the heart of miR-574$^{-/-}$ mice than in WT mice after TAC surgery (Fig EV2D and E). These findings suggest that miR-574$^{-/-}$ hearts are prone to cardiac remodeling and functional deterioration in response to β-adrenergic stimulation and pressure overload, which implies the potential cardioprotective function of miR-574.

## The therapeutic benefit of miR-574-5p/3p in cardiac remodeling using miRNA mimics

Having revealed a potential protective effect of either one or both of miR-574-5p and miR-574-3p strands in pathological cardiac remodeling, we sought to determine the therapeutic potential of these two miRNAs in the mouse heart *in vivo*. We used miRNA oligonucleotides that mimic endogenous miRNAs. We injected the mimetic oligomer of miR-574-5p, miR-574-3p, combined miR-574-5p/3p, or control miRNA (5 mg/Kg) into a tail vein of WT mice post-TAC surgery once a week for 4 weeks, following an established strategy (Fig 4A; Wang *et al*, 2016b). *In vivo* miR-574-5p and miR-574-3p levels were measured in the heart to ensure efficient delivery and miRNA stability (Fig EV3A). We first checked whether administration of miR-574-5p and miR-574-3p mimics cause organ toxicity and did not observe any obvious adverse side effects in the kidney and liver though both miRNA mimics were delivered systematically to the two organs among others in addition to the heart (Fig EV3B). TAC-induced cardiac hypertrophy and fibrosis were moderately reduced by miR-574-3p mimics, as indicated by HW/TL ratios, H&E and WGA staining, and picrosirius red staining (Fig 4A–C). miR-574-5p mimics were less effective compared to miR-574-3p in antagonizing cardiac fibrosis, and a combination of miR-574-5p and miR-574-3p showed modest synergistic effects, possibly due to the limited loading efficiency of both miR-574-5p and miR-574-3p into the RISC complex simultaneously in recipient cells. Expression of multiple hypertrophic and fibrotic marker genes was induced by the TAC surgery and reduced by miR-574-5p/3p treatment (Fig EV3C). Cardiac function was partially or moderately recovered with miR-574-3p and miR-574-5p treatment at the endpoint (Fig 4D and Table 2). In addition, miR-574-5p/3p treatment moderately decreased ROS production, restored ATP production and reduced cardiac apoptosis (though occurring at modest level under pressure overload) (Fig EV3D–F) compared to control miRNA mimics injection in the mice under TAC surgery. *In vitro* studies showed that both miR-574-5p and miR-574-3p reduced ISO-activated CM hypertrophy in primary mouse neonatal CM cells (Fig EV3G) but not CF cell proliferation in primary adult mouse CF cells (Fig EV3H). Intriguingly, miR-574-3p reduced α-SMA protein expression during TGF-β-triggered CF activation, while both miRNA strands significantly reduced α-SMA protein expression under the non-stimulation condition (Fig EV3I). Collectively, these results suggest that

sufficient miRNA mimics may drive the phenotypic rescue to reduce cardiac pathological remodeling by potentially targeting both CMs and CFs in HF mouse models.

## Transcriptome profiling identifies FAM210A as a common target of miR-574-5p/3p

To identify genes that are directly regulated by miR-574, we performed genome-wide RNA-Seq analyses with three pairs of P60 heart tissues from WT and miR-574$^{-/-}$ mice at baseline. Protein-coding genes that are significantly altered in expression between miR-574$^{-/-}$ and WT mice ($P_{adj} < 0.05$) were subject to Gene Ontology analyses. We identified 34 upregulated genes involved in the small molecule metabolism and the mitochondrial function (Fig 5A–C), and 49 downregulated genes in the electron transport chain (ETC) complex, oxidative phosphorylation, and tricarboxylic acid cycle (Fig 5A; Appendix Fig S2A and B, and Appendix Table S1).

Among the 34 upregulated genes, eight genes contain the predicted target seed sequence of miR-574-5p. *Fam210a* (family with sequence similarity 210 member A) is the only upregulated gene that contains the target seed sequence of both miR-574-5p and miR-574-3p (Fig 5D). We confirmed that the *Fam210a* transcript and protein level were indeed increased by ~50–60% in the heart of miR-574$^{-/-}$ mice (Fig 5E and F). Mouse *Fam210a* mRNA 3'UTR (untranslated region) contains two target sites for miR-574-5p and one target site of miR-574-3p (Fig 5G). Human *FAM210A* mRNA 3'UTR also contains the target seed sequence sites of miR-574-5p and miR-574-3p (Appendix Fig S2C). The dual luciferase reporter assay indicated that miR-574-5p/3p overexpression decreased the activity of the luciferase reporters containing each of the three wild-type seed sequence sites from mouse *Fam210a* mRNA 3'UTR (Fig 5H and I). This inhibitory effect of miR-574-5p/3p was abolished when the seed sequence was mutated in the 3'UTR of the reporter constructs (Fig 5H and I), suggesting that all these three target sites in the 3'UTR of *Fam210a* are direct binding sites for miR-574. All these results indicate that FAM210A is a common target of miR-574-5p and miR-574-3p *in vitro* and *in vivo*, and it may play a potential role in cardiac remodeling.

## FAM210A interacts with mitochondrial translation elongation factor and promotes mitochondrial protein expression

FAM210A was reported to be essential in maintaining skeletal muscle structure and strength (Tanaka *et al*, 2018; Tanaka *et al*, 2020). However, while it is a common downstream effector of both miR-574-5p and miR-574-3p, the function of FAM210A in the cardiac system and its molecular mechanism are unknown. Therefore, we sought to investigate the function of FAM210A. FAM210A is a transmembrane protein localized in the mitochondria (Zhang *et al*, 2008; He *et al*, 2012). FAM210A contains a mitochondrial targeting signal (MTS; cleavage site around Val[95], MitoCarta 2.0) peptide, a DUF1279 (Domain of Unknown Function 1279) domain containing a transmembrane (TM) domain (predicted by TOPCONS server (Tsirigos *et al*, 2015)), and a coiled coil (CC) at C-terminus (EMBL-EBI bioinformatics; Fig 6A, upper panel). The FAM210A gene is conserved in 213 organisms, including human, dog, cow, mouse, rat, chicken, frog, and zebrafish (Fig 6A, lower panel). RNA-Seq data show that *Fam210a* mRNA is ubiquitously expressed

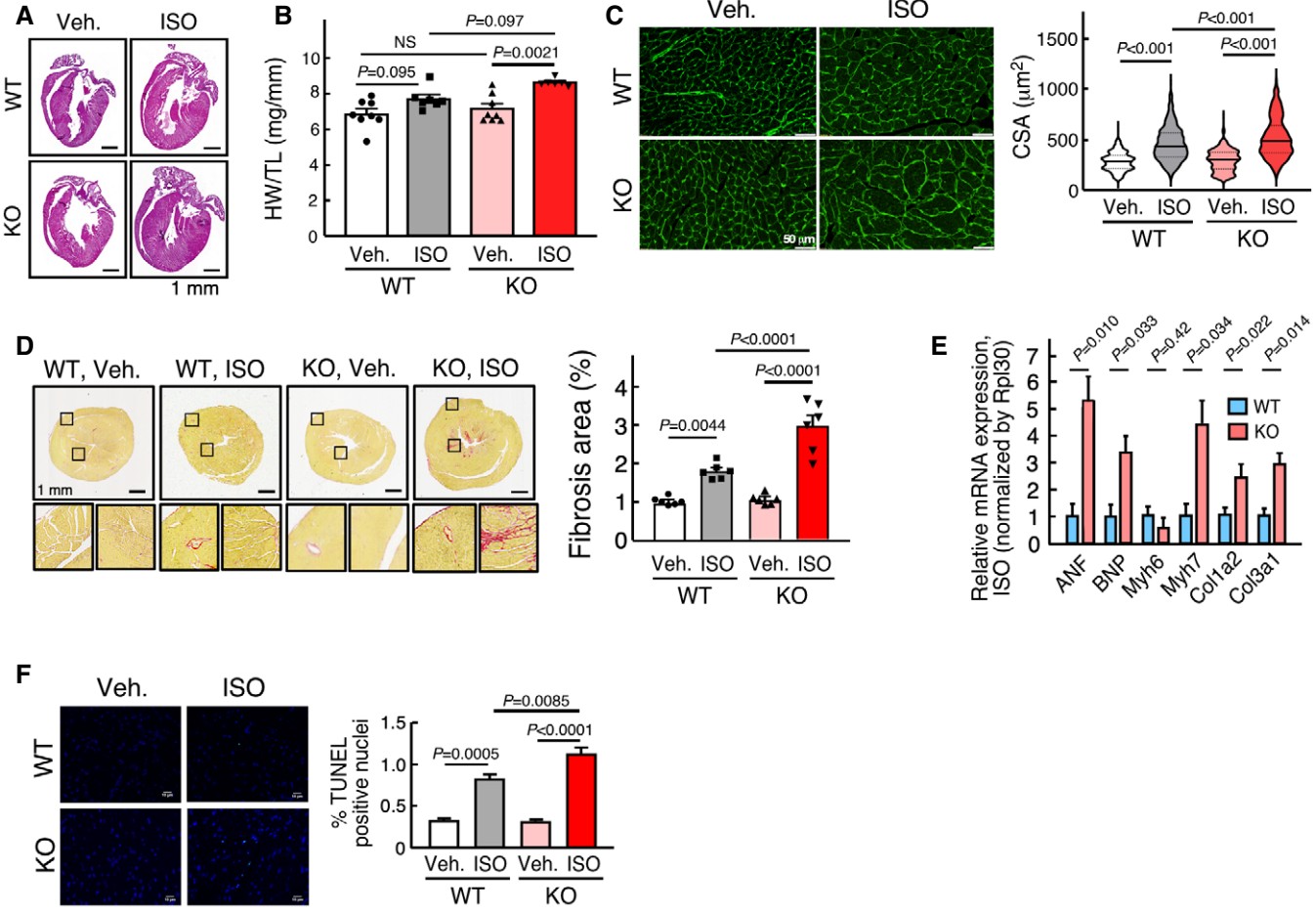

**Figure 2. miR-574 gene knockout augments ISO-induced pathological cardiac remodeling.**

A  H&E of hearts from WT and miR-574$^{-/-}$ mice with or without isoproterenol (ISO) treatment for 4 weeks. Mice were between 8–10 weeks old females.

B  Ratio of HW/TL (heart weight/tibia length) in WT and miR-574$^{-/-}$ mice (n = 8/7/8/6 for 4 groups of WT, Veh.; WT, ISO; KO, Veh.; KO, ISO).

C  WGA (wheat germ agglutinin) staining in WT and miR-574$^{-/-}$ mice. Cross-sectional area (CSA) of CMs was measured and quantified (n ≥ 500 cells). Scale bar: 50 μm. In the violin plot, black line shows median value for the group and peach dashed lines represent two quartile lines in each group.

D  Picrosirius red staining in WT and miR-574$^{-/-}$ mice (n = 6 per group). Scale bar: 1 mm.

E  RT–qPCR of fetal cardiac genes in WT and miR-574$^{-/-}$ mice with ISO treatment (n = 3 per group). ANF, atrial natriuretic factor; BNP, B-type natriuretic peptide; Myh6/Myh7, myosin heavy polypeptide 6/7; Col1a2/Col3a1, procollagen, type I, α2; type III, α1.

F  TUNEL assay for heart tissue sections from WT and miR-574$^{-/-}$ mice under ISO versus vehicle treatment (n = 6 per group). Scale bar: 10 μm.

Data information: Data were presented as mean ± SEM. P values were calculated by unpaired two-tailed Student t test (E), Kruskal–Wallis test with Dunn's multiple comparisons test (C), or two-way ANOVA with Tukey's multiple comparisons test (B, D, F).
Source data are available online for this figure.

in all organs, and is mostly enriched in the testis and heart (ventricle and atrium) of adult humans (Fig 6B, GTEx Portal; Consortium GT, 2013). The highest organ enrichment of FAM210A expression is consistent between human and mouse heart and skeletal muscle (Fig 6B and EV1A).

Global homozygous Fam210a$^{-/-}$ mice are embryonic lethal, while heterozygous Fam210a$^{+/-}$ mice are viable and fertile (Tanaka et al, 2018). LacZ reporter is highly expressed in the heart, brain, and skeletal muscle of Fam210a$^{+/-}$ mice at E12.5 (Appendix Fig S3A, IMPC phenotyping data) (Dickinson et al, 2016). Immunofluorescence and cellular fractionation followed by

immunoblotting revealed that endogenous or overexpressed FAM210A was mainly localized in the mitochondria of the mouse ACMs (Fig 6C), AC16 human CM cells (Fig 6D and Appendix Fig S3B), the murine heart (Fig 6E), and HEK293T human embryonic kidney cells (Appendix Fig S3C). Fractionation of mouse cardiac mitochondria showed that the majority of FAM210A was localized in mitoplasts (mitochondrial inner membrane and matrix), and a small fraction was located at the mitochondrial outer membrane (Fig 6F).

To determine the molecular function of FAM210A in the heart, we used a FAM210A-specific antibody to pull down the endogenous

FAM210A protein from crude mitochondria purified from the mouse hearts for mass spectrometry analysis of its interactome (Fig 7A). We used two stringent criteria to identify the strong binding partners of FAM210A, including (i) excluding the hits that were pulled down by IgG (#PSM of IgG> 0) and (ii) focusing on the proteins that are localized in the mitochondria, as FAM210A is mainly located in

the mitochondria (Fig 6). We found that mitochondrial elongation factor EF-Tu was one of the top ranking interacting mitochondrial proteins of FAM210A (Fig 7B; Appendix Table S2). Prior mass spectrometry screens also confirmed that FAM210A is the strongest binding protein of ATAD3A, which is tightly associated with the mitoribosome and EF-Tu (He *et al*, 2012).

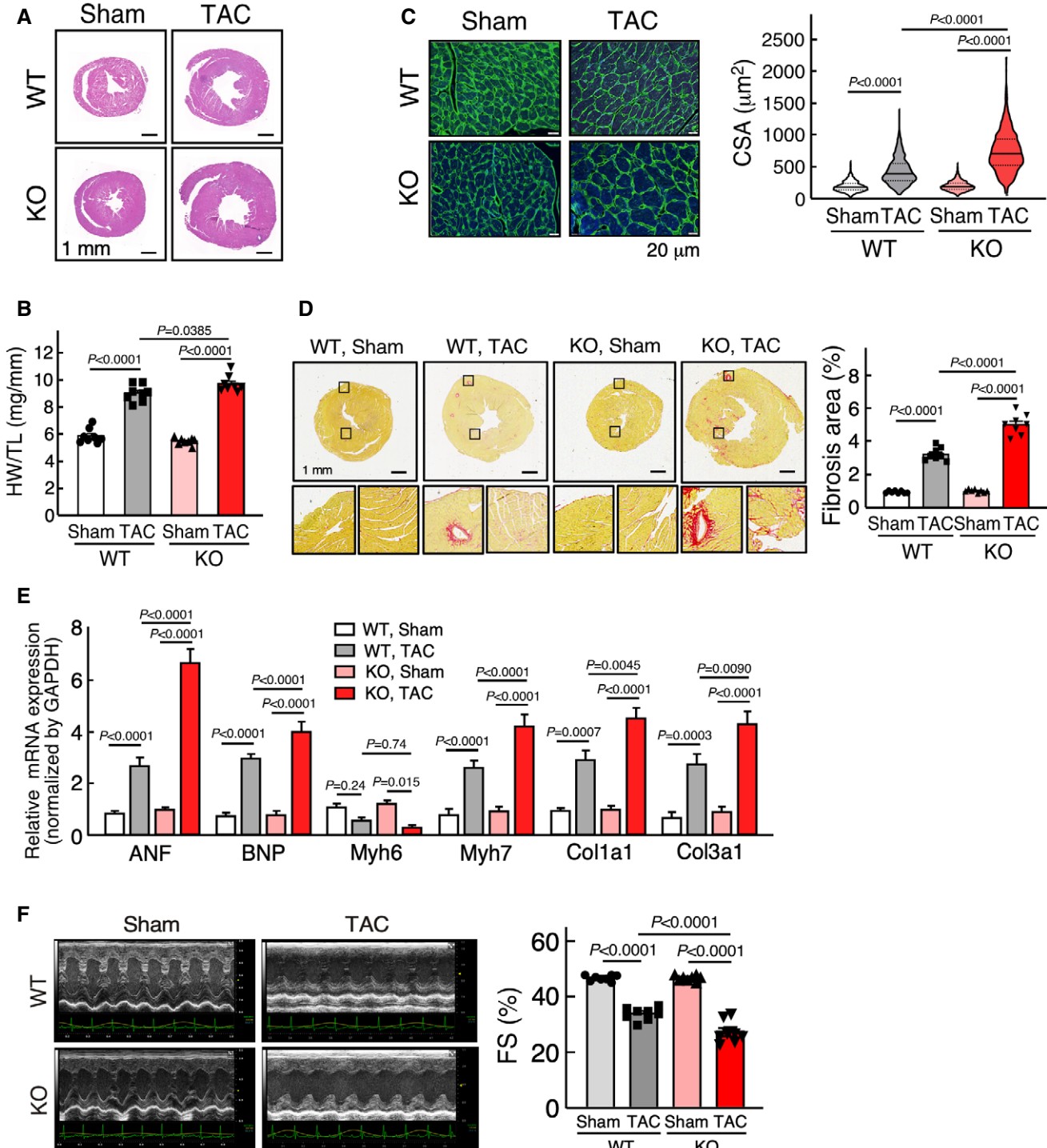

Figure 3.

◀

**Figure 3.   miR-574 gene knockout augments TAC-induced pathological cardiac remodeling.**

A   H&E staining of hearts from WT and miR-574$^{-/-}$ mice 4 weeks after transverse aortic constriction (TAC) surgery.
B   Ratio of HW/TL in WT and miR-574$^{-/-}$ mice (*n* = 8 per group).
C   WGA staining of hearts from WT and miR-574$^{-/-}$ mice 4 weeks after TAC surgery (*n* ≥ 500 cells). Scale bar: 20 µm. In the violin plot, black line shows median value for the group and peach dashed lines represent two quartile lines in each group.
D   Picrosirius red staining of hearts from WT and miR-574$^{-/-}$ mice under TAC surgery (*n* = 8 per group). Scale bar: 1 mm.
E   RT–qPCR of fetal cardiac genes in WT and miR-574$^{-/-}$ mice with TAC surgery (*n* = 4 per group).
F   Echocardiography measurement of cardiac functions of hearts from WT and miR-574$^{-/-}$ mice 4 weeks post-TAC (*n* = 8 per group). FS, fractional shortening.

Data information: Data were presented as mean ± SEM. *P* values were calculated by Kruskal–Wallis test with Dunn's multiple comparisons test (C) or two-way ANOVA with Tukey's multiple comparisons test (B, D-F).
Source data are available online for this figure.

**Table 1.   Cardiac function of WT and miR-574$^{-/-}$ mice under TAC surgery**

| Parameter | WT, Sham (*n* = 8) | | WT, TAC (*n* = 8) | | miR-574$^{-/-}$, Sham (*n* = 8) | | miR-574$^{-/-}$, TAC (*n* = 8) | |
|---|---|---|---|---|---|---|---|---|
| | BL | 4 week | BL | 4 week | BL | 4 week | BL | 4 week |
| HR (BPM) | 563.04 ± 7.02 | 582.30 ± 8.80 | 543.03 ± 7.80 | 558.54 ± 4.41 | 552.07 ± 8.01 | 564.70 ± 9.32 | 544.79 ± 9.78 | 547.87 ± 8.47 |
| LVID,s (mm) | 1.64 ± 0.161 | 2.04 ± 0.146 | 2.01 ± 0.065 | 3.02 ± 0.309* | 1.91 ± 0.107 | 1.86 ± 0.138 | 1.91 ± 0.088 | 3.75 ± 0.212$^{††}$ |
| LVID,d (mm) | 3.07 ± 0.278 | 3.27 ± 0.092 | 3.15 ± 0.096 | 3.85 ± 0.298 | 3.31 ± 0.108 | 3.18 ± 0.096 | 3.25 ± 0.084 | 4.91 ± 0.162$^{†}$ |
| LVV,s (µL) | 7.91 ± 0.41 | 8.98 ± 0.45 | 11.27 ± 0.79 | 25.48 ± 5.04*** | 8.17 ± 0.66 | 8.78 ± 1.30 | 14.13 ± 1.72 | 47.12 ± 3.04$^{†††}$ |
| LVV,d (µL) | 40.65 ± 2.04 | 41.58 ± 1.76 | 41.76 ± 1.97 | 55.62 ± 6.10** | 39.58 ± 2.90 | 41.02 ± 2.50 | 48.07 ± 2.90 | 77.47 ± 4.25$^{†††}$ |
| SV (µL) | 29.60 ± 2.934 | 32.64 ± 2.011 | 33.12 ± 2.478 | 28.47 ± 2.995 | 33.28 ± 2.260 | 32.17 ± 1.306 | 31.05 ± 1.511 | 26.73 ± 2.683 |
| EF (%) | 79.98 ± 0.97 | 79.44 ± 0.47 | 79.00 ± 0.67 | 57.50 ± 2.20*** | 78.50 ± 0.78 | 78.37 ± 0.49 | 78.90 ± 0.82 | 46.25 ± 1.68$^{††}$ |
| FS (%) | 45.46 ± 0.26 | 46.93 ± 0.36 | 45.00 ± 0.58 | 34.00 ± 0.80*** | 44.70 ± 0.33 | 46.75 ± 0.37 | 45.44 ± 0.35 | 26.50 ± 1.40$^{†††}$ |
| CO (mL/min) | 16.49 ± 1.633 | 18.77 ± 1.312 | 17.16 ± 1.003 | 16.00 ± 1.532 | 18.62 ± 1.225 | 17.47 ± 0.866 | 17.79 ± 1.013 | 15.61 ± 1.626 |
| LVM (mg) | 55.53 ± 5.318 | 58.2 ± 3.40 | 60.04 ± 3.403 | 120.82 ± 8.585*** | 61.42 ± 2.866 | 62.7 ± 4.120 | 60.26 ± 3.868 | 155.99 ± 11.734$^{††}$ |
| LVAWD,s (mm) | 1.05 ± 0.02 | 1.09 ± 0.03 | 0.90 ± 0.05 | 1.12 ± 0.06 | 1.06 ± 0.03 | 0.93 ± 0.06 | 0.83 ± 0.04 | 0.98 ± 0.06 |
| LVAWD,d (mm) | 0.67 ± 0.02 | 0.71 ± 0.02 | 0.77 ± 0.04 | 1.00 ± 0.05* | 0.74 ± 0.03 | 0.76 ± 0.04 | 0.68 ± 0.02 | 0.83 ± 0.05$^{†}$ |
| LVPWD,s (mm) | 0.87 ± 0.02 | 0.89 ± 0.02 | 0.88 ± 0.04 | 1.07 ± 0.04 | 0.86 ± 0.02 | 0.90 ± 0.04 | 0.84 ± 0.04 | 0.91 ± 0.06 |
| LVPWD,d (mm) | 0.55 ± 0.01 | 0.55 ± 0.01 | 0.70 ± 0.03 | 0.94 ± 0.03** | 0.54 ± 0.01 | 0.60 ± 0.05 | 0.68 ± 0.02 | 0.83 ± 0.05 |
| BW (g) | 25.60 ± 0.46 | 26.90 ± 0.52 | 26.02 ± 1.01 | 25.80 ± 0.74 | 25.90 ± 0.52 | 27.30 ± 0.70 | 26.06 ± 0.31 | 25.87 ± 0.50 |

BW, Body Weight (g); CO, Cardiac Output (ml/min); EF, Ejection Fraction (%); FS, Fractional Shortening (%); HR, Heart Rate (BPM, beats per minute); LVAWD,d, Left Ventricular Anterior Wall Diameter, diastole (mm); LVAWD,s, Left Ventricular Anterior Wall Diameter, systole (mm); LVID,d, Left Ventricular Internal Diameter, diastole (mm); LVID,s, Left Ventricular Internal Diameter, systole (mm); LVM, LV Mass (mg); LVPWD,d, Left Ventricular Posterior Wall Diameter, diastole (mm); LVPWD,s, Left Ventricular Posterior Wall Diameter, systole (mm); LVV,d, Left Ventricular Volume, diastole (µl); LVV,s, Left Ventricular Volume, systole (µl); SV, Stroke Volume (µl).
Values are expressed as mean ± SEM.
*$P$ < 0.05, **$P$ < 0.01, ***$P$ < 0.001 for WT, TAC vs. WT, Sham.
$^{†}P$ < 0.05, $^{††}P$ < 0.01, $^{†††}P$ < 0.001 for miR-574$^{-/-}$, TAC vs. WT, TAC.

To further validate that FAM210A is associated with EF-Tu in the mitochondria, we performed co-immunostaining for ectopically expressed C-terminal FLAG-tagged FAM210A and endogenous EF-Tu in a human AC16 ventricular cardiomyocyte cell line. Double immunostaining and two-color histogram analysis revealed significant co-localization between FAM210A and EF-Tu in the mitochondria (Fig 7C). IP and IB for endogenous FAM210A confirmed that EF-Tu was a bona fide interacting protein of FAM210A in the murine heart (Fig 7D, left panel). We further confirmed the interaction between EF-Tu and FAM210A by overexpression of FLAG-tagged FAM210A (Fig 7D, right panel). We mapped the interacting domain of FAM210A with EF-Tu using a series of truncated FAM210A mutants (Fig 7E). We found that the DUF1279 domain was responsible for binding EF-Tu, but the C-terminal CC domain was not required for binding. Instead, the deletion of the CC domain

enhances the interaction between FAM210A and EF-Tu. This suggests that the CC domain might function as a regulatory module to influence the interaction with EF-Tu. The deletion of the single TM domain led to deficient expression of the truncated protein, possibly due to mis-localization and subsequent degradation of the mutant FAM210A protein.

Since FAM210A interacts with mitochondrial translation elongation factor EF-Tu, we hypothesize that FAM210A may participate in regulating the protein expression of mitochondrial-encoded genes. We measured the MEG and NEMG protein expression after overexpression or knockdown of FAM210A in human CM cell line. We found that overexpression or knockdown of FAM210A increased or decreased MEG protein expression, respectively, but did not affect NEMG protein expression significantly (Fig 7F and G; Appendix Fig S4A and B), suggesting that FAM210A promotes the

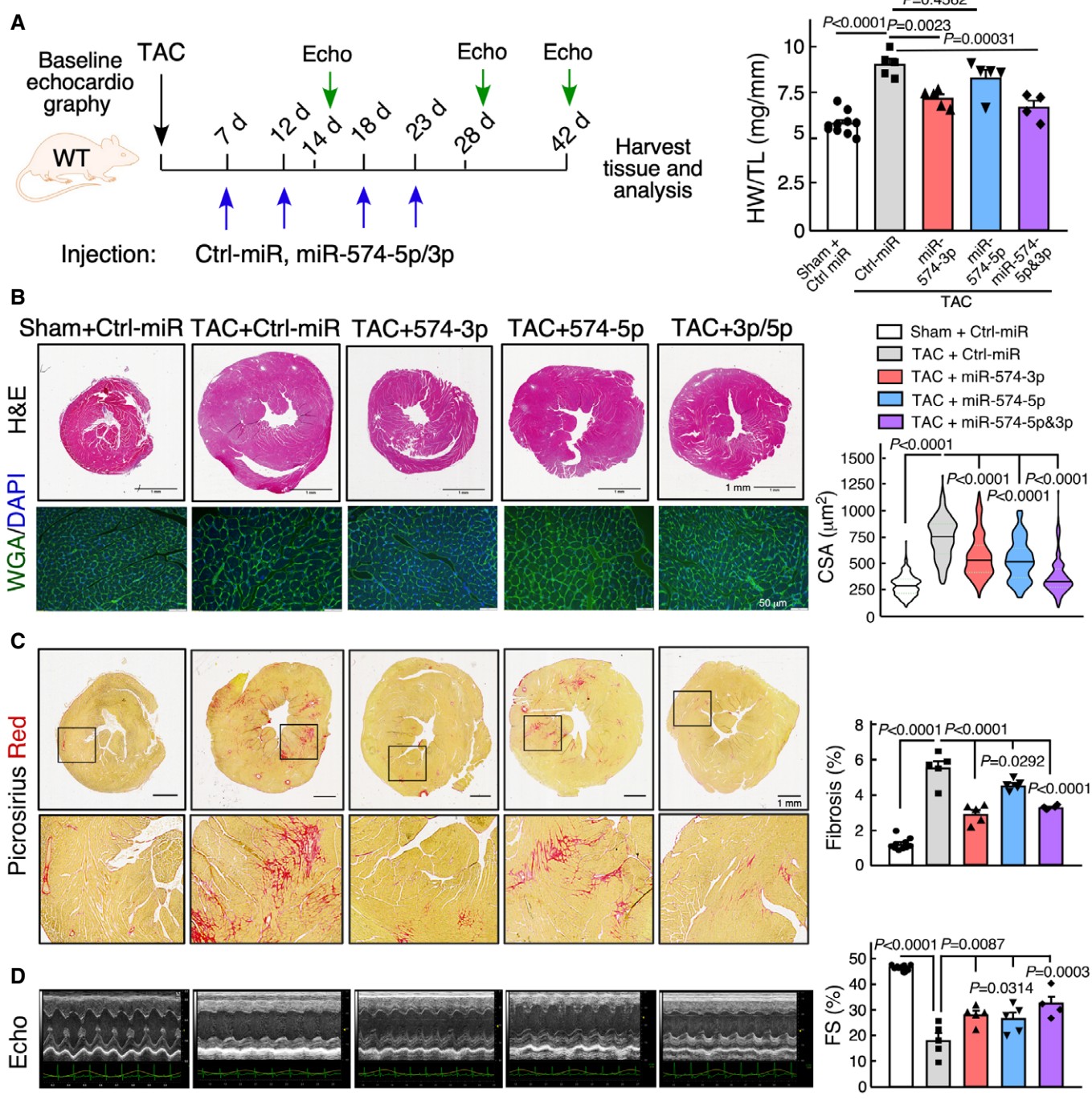

**Figure 4. The therapeutic benefit of miR-574-5p/3p against cardiac pathological remodeling.**

A   The schematic of therapeutic treatment of the TAC model using miRNA mimics followed by phenotypic characterizations. HW/TL ratio is measured ($n$ = 4–10 per group). Ctrl-miR: negative control miRNA mimics.

B   H&E and WGA staining of murine hearts in the therapeutic models. Scale bar: 1 mm for H&E and 50 μm for WGA. Cardiomyocyte surface area was quantified from WGA staining in the right panel. >120 CMs are quantified in each group. The black line shows medium value for the group and the dotted lines represent two quartile lines in each group.

C   Picrosirius red staining of murine hearts in the therapeutic models. Scale bar: 1 mm. The fibrotic area was quantified in the right panel ($n$ = 4–10 per group).

D   Echocardiography analysis of the cardiac function of mice in the therapeutic models at the endpoint of 6 weeks post-surgery ($n$ = 4–10 per group).

Data information: Data were presented as mean ± SEM. $P$ values were calculated by Kruskal–Wallis test with Dunn's multiple comparisons test (A) or one-way ANOVA with Tukey's multiple comparisons test (B-D).
Source data are available online for this figure.

**Table 2. Cardiac function of WT mice with miRNA treatment under TAC surgery**

| Parameter | Sham, Ctrl-miR (n = 10) | | TAC, Ctrl-miR (n = 5) | | TAC, miR-574-3p (n = 5) | | TAC, miR-574-5p (n = 5) | | TAC, miR-574-5p & 3p (n = 4) | |
|---|---|---|---|---|---|---|---|---|---|---|
| | BL | 6 week | BL | 6 week | BL | 6 week | BL | 6 week | BL | 6 week |
| HR (BPM) | 565.04 ± 7.02 | 607.32 ± 7.96 | 571.05 ± 4.92 | 532.03 ± 9.32*** | 540.66 ± 16.30 | 553.35 ± 7.92 | 545.87 ± 14.16 | 563.16 ± 14.73 | 550.73 ± 11.49 | 549.72 ± 11.70 |
| LVID,s (mm) | 1.92 ± 0.039 | 2.00 ± 0.064 | 2.10 ± 0.112 | 3.90 ± 0.33*** | 2.28 ± 0.12 | 2.81 ± 0.107†† | 2.24 ± 0.081 | 2.96 ± 0.298 | 1.88 ± 0.083 | 2.47 ± 0.166††† |
| LVID,d (mm) | 3.29 ± 0.097 | 3.51 ± 0.094 | 3.37 ± 0.121 | 4.59 ± 0.19*** | 3.42 ± 0.09 | 3.92 ± 0.076† | 3.52 ± 0.037 | 3.95 ± 0.220 | 3.25 ± 0.087 | 3.66 ± 0.163††† |
| LVV,s (µL) | 7.86 ± 0.330 | 8.99 ± 0.400 | 14.68 ± 1.68 | 66.21 ± 6.36*** | 18.08 ± 2.54 | 26.13 ± 4.08†† | 17.16 ± 1.49 | 27.91 ± 3.39†† | 11.14 ± 1.19 | 25.59 ± 6.46††† |
| LVV,d (µL) | 40.34 ± 1.63 | 43.58 ± 1.58 | 46.74 ± 3.58 | 100.38 ± 5.63*** | 48.44 ± 3.01 | 67.98 ± 5.96† | 51.59 ± 1.28 | 56.13 ± 8.45††† | 43.07 ± 2.59 | 62.48 ± 8.36†† |
| SV (µL) | 36.83 ± 0.395 | 37.00 ± 0.447 | 32.06 ± 3.070 | 28.95 ± 5.27 | 30.36 ± 4.06 | 36.67 ± 2.404† | 34.43 ± 0.796 | 33.36 ± 2.249 | 31.66 ± 2.225 | 34.76 ± 3.898 |
| EF (%) | 80.52 ± 0.26 | 79.45 ± 0.31 | 68.49 ± 2.96 | 33.32 ± 3.25*** | 62.33 ± 5.71 | 61.88 ± 3.36†† | 66.87 ± 2.30 | 54.63 ± 8.25†† | 73.67 ± 2.09 | 62.06 ± 5.08††† |
| FS (%) | 47.93 ± 0.27 | 46.95 ± 0.29 | 48.10 ± 0.42 | 17.46 ± 1.91*** | 46.58 ± 0.31 | 34.27 ± 3.77†† | 47.18 ± 0.50 | 30.44 ± 2.25† | 45.40 ± 0.39 | 35.05 ± 2.06††† |
| CO (mL/min) | 20.95 ± 0.370 | 21.39 ± 0.422 | 18.32 ± 1.756 | 15.49 ± 2.73* | 16.41 ± 2.23 | 20.02 ± 1.908 | 18.81 ± 0.845 | 18.38 ± 1.221 | 17.51 ± 1.444 | 18.58 ± 1.905 |
| LVM (mg) | 54.98 ± 0.518 | 56.10 ± 1.336 | 60.26 ± 2.871 | 153.28 ± 20.31*** | 65.95 ± 2.35 | 96.17 ± 4.521†† | 59.91 ± 4.109 | 110.83 ± 11.313† | 62.64 ± 2.777 | 99.49 ± 8.788†† |
| LVAWD,s (mm) | 1.03 ± 0.02 | 1.09 ± 0.03 | 0.80 ± 0.02 | 1.14 ± 0.04 | 0.77 ± 0.01 | 0.95 ± 0.05† | 0.74 ± 0.04 | 0.97 ± 0.09† | 0.94 ± 0.05 | 0.92 ± 0.09 |
| LVAWD,d (mm) | 0.67 ± 0.01 | 0.71 ± 0.02 | 0.60 ± 0.03 | 1.22 ± 0.07** | 0.66 ± 0.02 | 0.79 ± 0.04†† | 0.59 ± 0.03 | 0.82 ± 0.07† | 0.72 ± 0.02 | 0.80 ± 0.06† |
| LVPWD,s (mm) | 0.88 ± 0.02 | 0.89 ± 0.02 | 0.83 ± 0.04 | 1.00 ± 0.07 | 0.77 ± 0.02 | 1.08 ± 0.06 | 0.72 ± 0.04 | 0.89 ± 0.08 | 0.92 ± 0.02 | 0.86 ± 0.07 |
| LVPWD,d (mm) | 0.55 ± 0.02 | 0.55 ± 0.01 | 0.60 ± 0.03 | 0.92 ± 0.05** | 0.60 ± 0.02 | 0.81 ± 0.04† | 0.54 ± 0.03 | 0.80 ± 0.08† | 0.58 ± 0.09 | 0.77 ± 0.03†† |
| BW (g) | 26.10 ± 0.25 | 28.49 ± 0.47 | 26.74 ± 0.11 | 25.26 ± 0.99* | 27.30 ± 0.45 | 26.60 ± 1.05 | 26.90 ± 0.24 | 26.77 ± 0.93 | 26.70 ± 0.76 | 27.08 ± 0.36 |

Values are expressed as mean ± SEM.

*$P < 0.05$, **$P < 0.01$, ***$P < 0.001$ for TAC, Ctrl-miR vs. Sham, Ctrl-miR.

†$P < 0.05$, ††$P < 0.01$, †††$P < 0.001$ for TAC, Ctrl-miR-574-3p/5p/3p & 5p vs. Sham, Ctrl-miR.

expression of MEGs but not NEMGs. In summary, FAM210A modulates the expression of multiple MEGs encoding ETC component proteins, possibly by interacting with EF-Tu.

## miR-574 modulates mitochondrial-encoded protein expression via regulation of FAM210A and influences mitochondrial activity

We have shown that Fam210a is a direct target of miR-574 and FAM210A regulates mitochondrial-encoded protein expression. Therefore, we sought to determine whether miR-574 regulates mitochondrial-encoded protein expression via regulation of FAM210A. We first confirmed that overexpression of miR-574-5p/3p reduced *FAM210A* mRNA expression while inactivation of miR-574-5p/3p by anti-miRNA inhibitors increased *FAM210A* mRNA expression (Fig 8A and B). We then measured the MEG and NEMG protein expression with transfection of miR-574-5p, miR-574-3p, or the anti-miR inhibitor of each. Overexpression of either miR-574-5p or miR-574-3p reduced MEG protein expression (Figs 8C and Fig EV4A), while transfection of anti-miR-574-5p or anti-miR-574-3p inhibitor increased MEG protein expression (Figs 8D; Fig EV4B). In contrast, NEMG protein expression was not significantly affected in both experiments. These observations suggest that miR-574 downregulates MEG protein expression by targeting FAM210A.

To examine whether miR-574 or miR-574-FAM210A axis plays a protective role in ISO-triggered increase of CM hypertrophy and decrease of mitochondrial activity, we measured the hypertrophy and mitochondrial phenotypes in the same set of treated cells in the presence or absence of ISO stimulation. Transfection of miR-574-5p or miR-574-3p mimics antagonized ISO-induced hypertrophy of AC16 CMs (Fig EV4C) while transfection of anti-miR-574-5p or anti-miR-574-3p inhibitors exaggerated ISO-induced CM cell hypertrophy (Fig EV4D). Also, miR-574-5p or miR-574-3p mimics partially

restored the reduced mitochondrial membrane potential (Figs 8E and Fig EV4E) and ATP production in AC16 cells (Fig 8F). To confirm the mitochondrial protective role of miR-574 in cardiac myocytes, we further characterized the effects on mitochondrial function and morphology in CMs of WT and miR-574 null mice. ISO treatment triggered more impaired membrane potential (Fig 8G) and reduced ATP production in isolated primary CMs from the heart of miR-574$^{-/-}$ mice than those from WT hearts (Fig 8H). In line with disrupted mitochondrial functions in the cultured CM cells *in vitro*, we observed more mitochondria swelling, compromised cristae formation, and overall mitochondrial disarray in miR-574$^{-/-}$ hearts than WT hearts upon ISO stress using electron microscopy (Fig EV4F and G). Moreover, co-overexpression of FAM210A in the presence of miR-574-5p/3p mimics transfection rescued MEG expression without affecting NEMG protein expression or mitochondria copy number (Figs 8I and Fig EV4H and I), further confirming the role of miR-574-FAM210A axis in regulating MEG protein expression in human CM cells.

To determine whether loss of miR-574 causes dysregulation of MEG and NEMG expression for the ETC complexes and compromises ETC activity *in vivo*, we first quantitatively measured their protein expression levels and found that FAM210A and MEG protein expression was significantly induced in miR-574 KO hearts compared to WT hearts at 3 days after TAC surgery (Figs 8J and Fig EV5A and B). In contrast, NEMG protein expression was not altered (Figs 8J, Fig EV5A and B). Secondly, we measured the mitochondrial DNA content and found no obvious changes (Fig EV5C). Thirdly, we also measured the enzymatic activity of multiple ETC complexes using the isolated cardiac mitochondria from WT and miR-574 null mice 4 weeks post-TAC surgery. We found that the activity of complexes I, III and IV (but not complex II) was more significantly reduced in miR-574$^{-/-}$ hearts compared to WT hearts

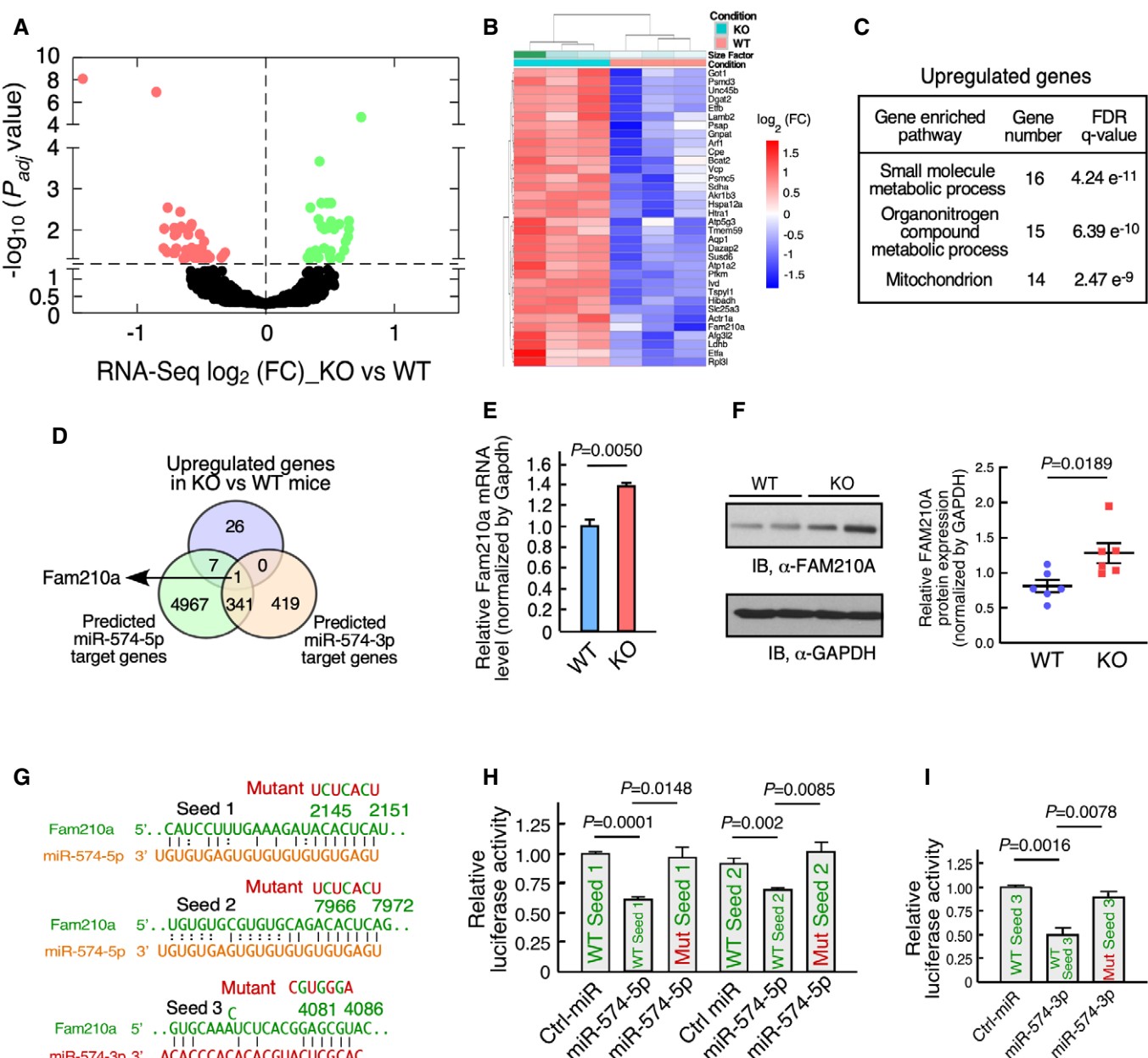

**Figure 5. Identification of Fam210a as a common target of miR-574-5p/3p *in vivo*.**

A   Volcano curve analysis of dysregulated genes in miR-574$^{-/-}$ versus WT heart. The $P_{adj}$ is the $P$ values adjusted for multiple testing with correction using the Benjamini–Hochberg procedure. The dots above the dashed line show $P_{adj} < 0.05$.

B   Heatmap of significantly upregulated genes in miR-574$^{-/-}$ mice at baseline analyzed by RNA-Seq. P60 male mice, $n = 3$ per group, $P_{adj} < 0.05$. The $P_{adj}$ is the $P$ values adjusted for multiple testing with correction using the Benjamini–Hochberg procedure.

C   Gene Ontology analysis of enriched pathways of upregulated genes in RNA-Seq. The top three pathways are listed with enriched gene sets.

D   Identification of Fam210a as a target of miR-574-5p and miR-574-3p. Seed sequence-bearing target genes were predicted by TargetScan.

E   Expression of *Fam210a* mRNA in the heart of miR-574$^{-/-}$ and WT mice from 3 biological replicates.

F   Expression of FAM210A protein in the heart of miR-574$^{-/-}$ and WT mice. Protein intensity was quantified in the right panel ($n = 6$ per group).

G   Seed sequences of miR-574-5p and miR-574-3p in *Fam210a* mRNA 3'UTR (TargetScan) and their mutants.

H   Dual luciferase reporter assays with co-transfection of miR-574-5p mimics and FLuc-Fam210a 3'UTR bearing wild-type and mutant seed sequences. The assays were performed in three biological replicates.

I   Dual luciferase reporter assays with co-transfection of miR-574-3p mimics and FLuc-Fam210a 3'UTR bearing wild-type and mutant seed sequences. The assays were performed in three biological replicates.

Data information: Data were presented as mean ± SEM. $P$ values were calculated by unpaired two-tailed Student $t$ test (E, F, H, I).

(Fig EV5D). These observations suggest that the miR-574-FAM210A axis prevents excessive expression of MEG proteins in the ETC complex, thereby possibly maintaining mitochondrial protein homeostasis and normal mitochondrial functions. This mechanism may contribute to the cardioprotective effects of miR-574 in the mouse HF models.

## Discussion

This study was initiated from our data mining of unbiased screenings of heart disease-related miRNAs in multiple independent studies (Fig 1). We found that miR-574-5p and miR-574-3p were induced in human and mouse diseased hearts. We demonstrated

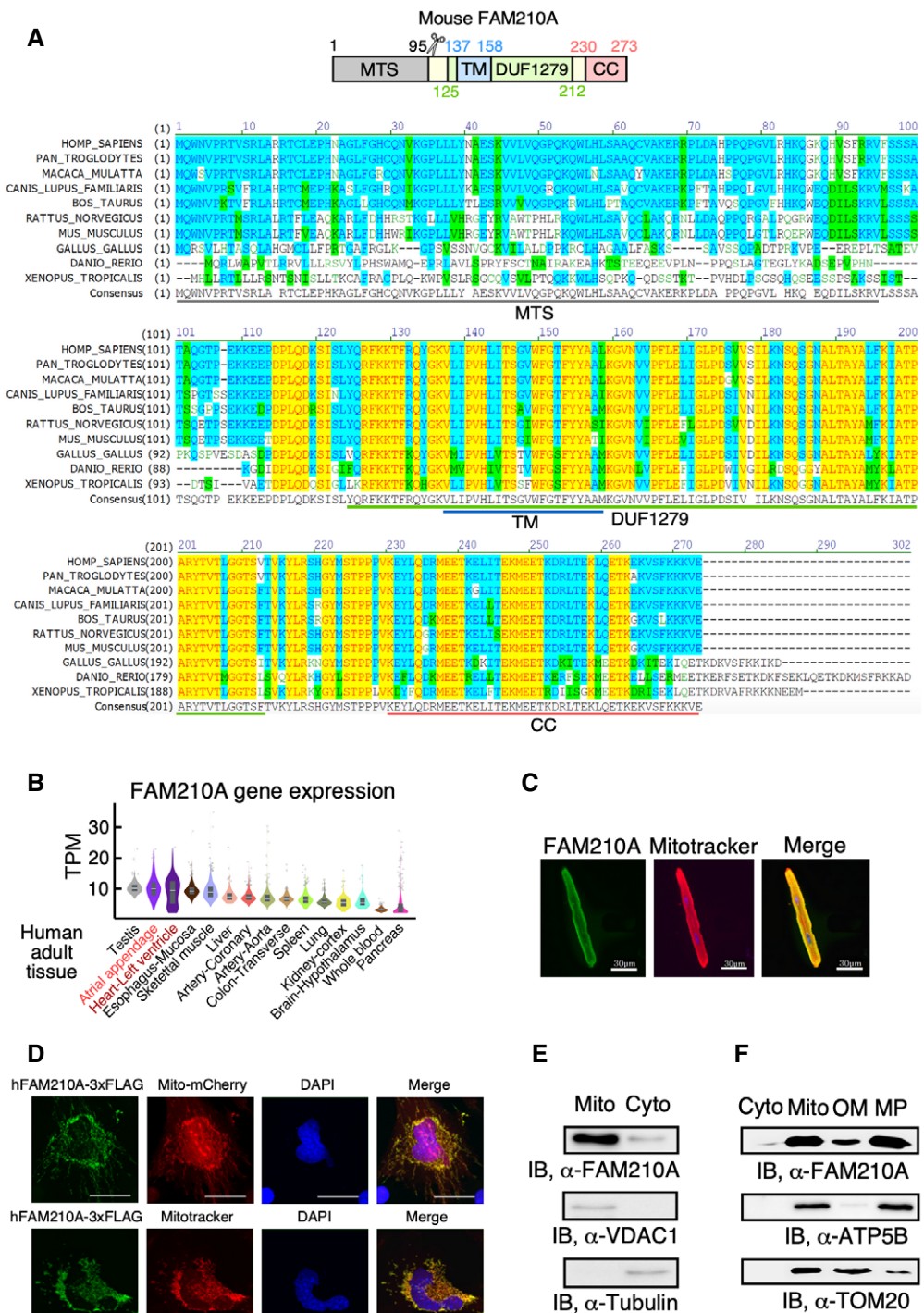

**Figure 6.**

**Figure 6. FAM210A is a conserved mitochondrial protein located in the mitoplast.**

A The schematic of FAM210A protein domain composition and evolutionary conservation of FAM210A protein sequence from 10 representative species analyzed by AlignX software. The yellow color shows completely conserved residues across all the species at a given position; the blue color indicates consensus residues derived from a block of similar residues at a given position, while the green color stands for the consensus residues derived from the occurrence of greater than 50% of a single residue at a given position. The green letters suggests weak similarity to consensus residues at a given position.

B The expression level of *FAM210A* mRNA in 15 human organs from the GTEx Portal database. TPM: transcripts per million of reads. The central band shows the median value, and the box indicates the first and third quartiles. The sample size of human samples used for quantification in GTEx Portal database (from testis to pancreas): $n = 361/429/432/555/803/240/432/226/406/578/241/202/85/328/755$.

C Cellular localization of FAM210A in primary mouse ACMs detected by IF. Scale bar: 30 μm.

D Cellular localization of recombinant FAM210A in human AC16 CM cells. FAM210A-3xFlag expression plasmid was transfected into AC16 cells for 48 h. Scale bar: 40 μm.

E Immunoblot detection of FAM210A in isolated cytoplasmic and mitochondrial fractions from mouse hearts. VDAC1 and tubulin are used as fraction controls. Cyto: cytoplasm; Mito: mitochondria. A representative image is shown from three biological replicates.

F Sub-organelle localization of FAM210A in isolated mouse cardiac mitochondria. TOM20 and ATP5B are used as the outer membrane (OM) and mitoplast (MP) markers, respectively.

Source data are available online for this figure.

that miR-574-5p/3p play cardioprotective roles using a miR-574 genetic knockout mouse model and intravenous injection of miR-574-5p/3p mimics in mouse HF models. RNA-Seq of baseline miR-574-null mouse hearts revealed *Fam210a* mRNA as a critical downstream target of both strands of miR-574. Biochemical analyses suggest that FAM210A functions in regulating mitochondrial-encoded protein expression. Furthermore, we show that mitochondrial-encoded protein expression is dysregulated followed by compromised mitochondrial function in miR-574-null mouse hearts upon cardiac stresses. Therefore, miR-574 restricts FAM210A expression and may act as a molecular brake to maintain mitochondrial homeostasis and normal cardiac function (Fig 8K).

The guide and passenger strands of miR-574 are both loaded onto the RISC and functionally active (Zhang *et al*, 2014a; Yao *et al*, 2017). miR-574-5p has been identified as an oncogenic miRNA (Meyers-Needham *et al*, 2012) and is involved in neurogenesis (Zhang *et al*, 2014a). miR-574-3p is recognized as an anti-tumor miRNA (Ishikawa *et al*, 2014; Yao *et al*, 2017). Using a loss-of-function model, we show that miR-574$^{-/-}$ mice display more severe cardiac hypertrophy, mitochondrial dysfunction, and fibrosis compared to WT mice after ISO injection and TAC surgery, suggesting a cardioprotective function of miR-574. To confirm the in-trans effects of miR-574 in cardioprotection, we injected miR-574-5p/3p mimics into miR-574$^{-/-}$ mice and could significantly rescue the cardiac disease phenotype under ISO treatment including reduced CM hypertrophy, fibrosis (accompanied by reduced hypertrophy and fibrosis marker gene expression), and myocyte death (Appendix Fig S5). miR-574-5p may play a more important role in regulating CM hypertrophy than miR-574-3p since we observed more robust induction of this guide strand miRNA in CMs *in vivo* (Fig EV1C) and *in vitro* (Fig EV1F). Importantly, both miR-574-5p and miR-574-3p can inhibit ISO-induced CM hypertrophy in primary mouse neonatal CMs and human AC16 CM cells (Figs EV3G and EV4C and D). On the other hand, *in vitro* studies suggest that neither miR-574-5p nor miR-574-3p antagonizes CF proliferation (Fig EV3H). However, miR-574-3p shows inhibitory effects in TGF-β-mediated CF activation in primary mouse CF cells while both miRNA strands inhibited α-SMA protein expression at baseline (Fig EV3I). These results suggest that exogenous miR-574-5p and miR-574-3p mimics may primarily target CMs as a synergistic pair and could also

target CFs as a complementary mechanism in the rescue and therapeutic models (Appendix Fig S5 and Fig 4).

In the therapeutic model, miR-574-3p mimics reduced cardiac hypertrophy to a slightly less extent but antagonize cardiac fibrosis to a higher extent than miR-574-5p mimics (Fig 4). We think that there might be two explanations: 1) The loading efficiency of miR-574-5p and miR-574-3p onto RISC with Ago2 may be different after miRNAs entering CM and CF cells via nanoparticle delivery; 2) The function and contribution of full spectrum of individual targets of miR-574-5p and miR-574-3p may play a differential role in cardiac remodeling (e.g., in hypertrophy versus fibrosis). Consistent with our *in vivo* findings, we found that miR-574-5p showed stronger anti-hypertrophic activity but weaker inhibition of cardiac fibroblast activation compared to miR-574-3p in primary CM and CF cell cultures (Fig EV3G–I). We cannot rule out the possibility of indirect cardioprotective effects of systematic delivery of miR-574-5p and miR-574-3p mimics in other organs. At least, we did not observe severe side effects or toxicity in the liver and kidney (Fig EV3B).

Based on RNA-Seq of hearts from WT and miR-574 null mice followed by a series of biochemical assays, we validated FAM210A as a bona fide miR-574-5p/3p target in human AC16 CM cell line *in vitro* (Figs 8A–D and Fig EV4A and B) and in murine hearts *in vivo* (Figs 8J and Fig EV5). Using human AC16 CM cell line culture system, we showed that miR-574 regulates MEG protein expression by targeting FAM210A after manipulating the ratio of miR-574 and FAM210A in an acute phase (Figs 8I and Fig EV4H). We also showed that loss of miR-574 leads to increase in FAM210A and MEG (but not NEMG) protein expression in the hearts of miR-574 null mice upon pressure overload stress *in vivo* (Figs 8J and Fig EV5A–C) and reduced mitochondrial potential and ATP production in primary miR-574 null CMs (Fig 8G and H). This imbalance between MEG and NEMG protein expression in the early stage may cause ETC dysfunction, mitochondrial stress, and more severe cardiac dysfunction in the late stage as we observed (Fig 3 and Figs EV5D). Overexpression of miR-574-5p or miR-574-3p in AC16 human CM cells partially reverses ISO-induced mitochondrial dysfunction (Fig 8E and F). Therefore, miR-574-5p and miR-574-3p may function as potential checkpoint molecules to prevent excessive MEG protein expression (overcompensation), maintain

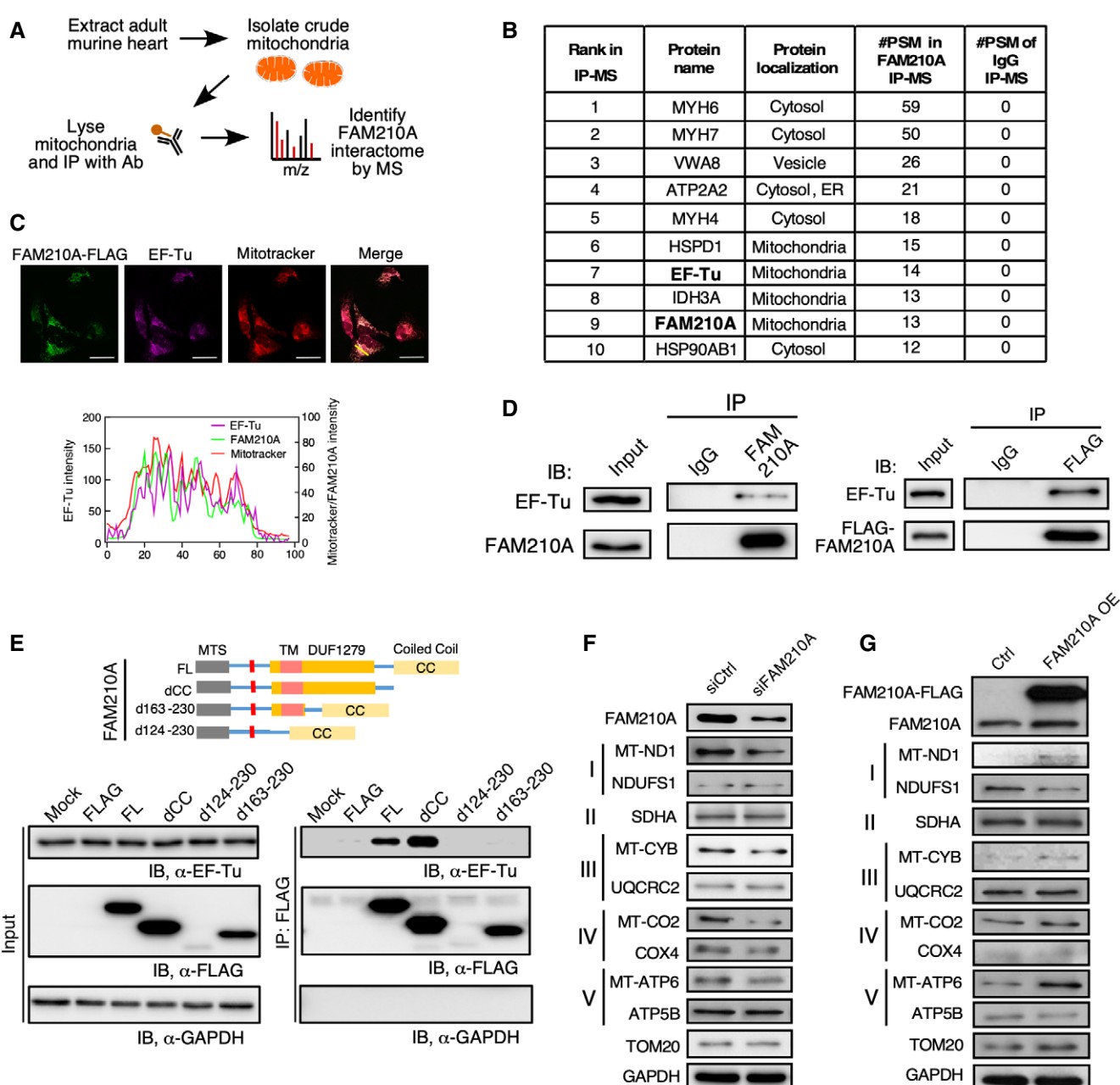

**Figure 7. FAM210A interacts with EF-Tu and regulates mitochondrial-encoded protein expression.**

A   Antibody-based affinity purification of endogenous FAM210A from purified murine cardiac mitochondria coupled with mass spectrometry analysis.

B   Top list of FAM210A-interacting proteins in IP-MS (immunoprecipitation-mass spectrometry) analysis. PSM: peptide spectrum match.

C   Confocal imaging of co-localization of FAM210A-3xFLAG and endogenous EF-Tu by immunostaining. Scale bar: 40 μm.

D   Left panel: IP-IB validates FAM210A-interacting proteins using pre-immune IgG and FAM210A antibodies. Right panel: IP-IB validates FAM210A-interacting proteins in HEK293T cell lysates subjected to IP using pre-immune IgG and anti-FLAG antibodies. A representative image is shown in three biological replicates.

E   Mapping of interacting domains in FAM210A for binding EF-Tu.

F, G   Protein expression of mitochondrial ETC component genes was measured in AC16 cells after following treatments in comparison with control cells: (F) siRNA knockdown of FAM210A. (G) FAM210A overexpression. Quantitative data were shown in Appendix Fig S4A and B.

Source data are available online for this figure.

mitochondrial homeostasis, and reduce CM hypertrophy (Fig 8K). On the other hand, both miRNA strands especially miR-574-3p may directly inhibit CF activation in combination with non-autonomous anti-fibrotic effects from crosstalk between CMs and CFs, which needs to be further studied in cardiac cell type-specific miR-574 conditional knockout models in the future.

A genome-wide meta-analysis shows that the genetic variations near FAM210A are associated with reduced lean mass and increased risk of bone fracture related to its function in maintaining normal skeletal muscle functions (Estrada *et al*, 2012; Tanaka *et al*, 2018). In this study, we discovered that FAM210A regulates MEG protein expression. FAM210A is ubiquitously expressed in multiple cell types (CMs, CFs, skeletal muscle cells, *etc.*) and organs (heart, skeletal muscle, brain, *etc.*; Tanaka *et al*, 2018; Figs 6B and Fig EV1A), implying a conserved and generalized role for this protein. Similar to FAM210A, miR-574 is ubiquitously expressed and more enriched in heart, brain, and skeletal muscle (Fig EV1A). Thus, the miR-574-mediated regulation of FAM210A may also play

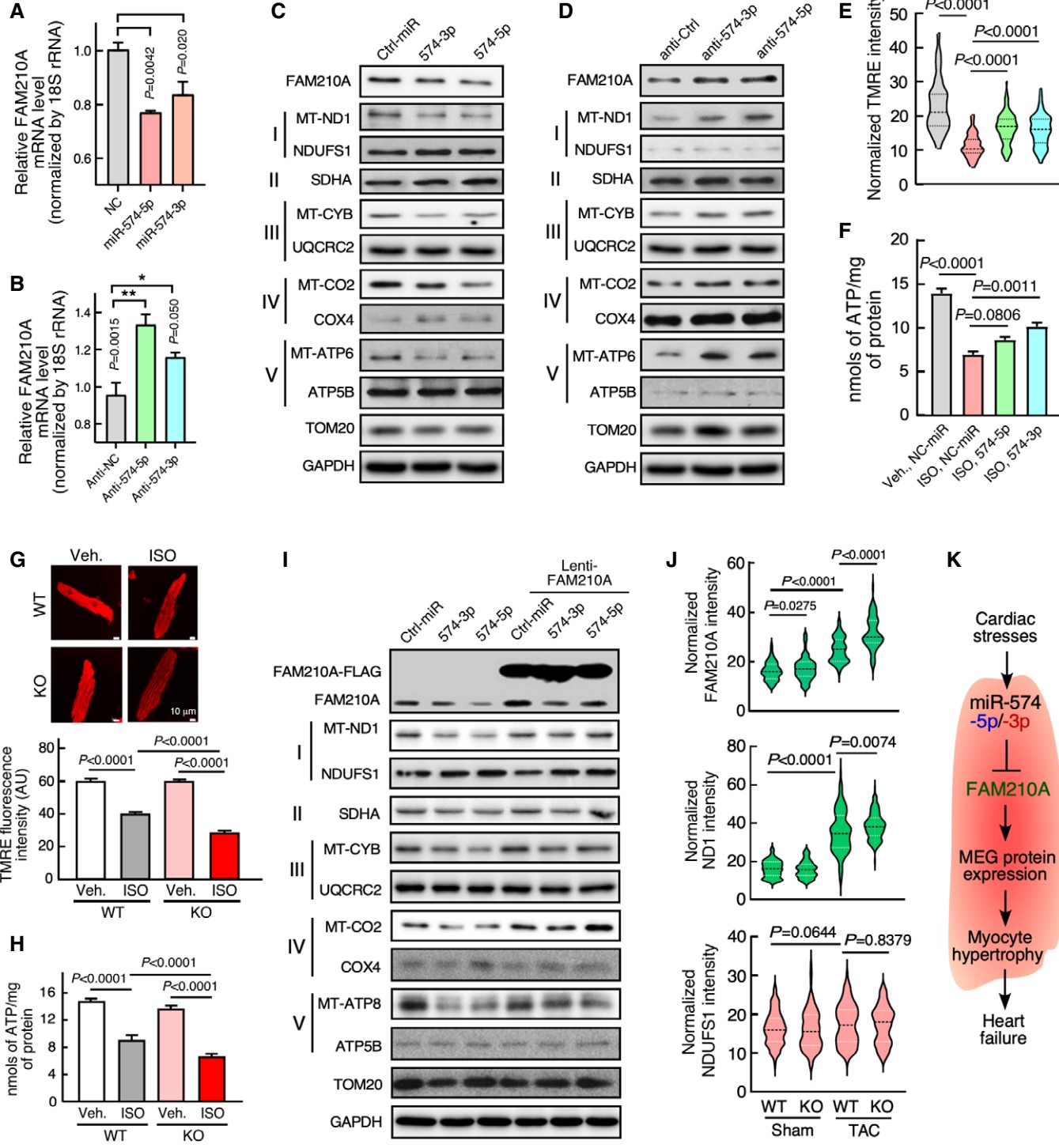

**Figure 8.**

**Figure 8. miR-574 regulates mitochondrial protein expression through targeting FAM210A.**

A       miR-574-5p and miR-574-3p overexpression reduced *FAM210A* mRNA expression in AC16 cells (*n* = 3 in each group).
B       miR-574-5p and miR-574-3p inhibition increased *FAM210A* mRNA expression in AC16 cells (*n* = 3 in each group).
C, D    Protein expression of mitochondrial ETC component genes was measured in AC16 cells after following treatments in comparison with control cells: (C) miR-574-5p or miR-574-3p overexpression. (D) Transfection of anti-miR-574-5p or anti-miR-574-3p inhibitor.
E, F    Mitochondrial membrane potential and ATP production in AC16 cells under ISO treatment (10 μM for 24 h) with overexpression of miR-574-5p or miR-574-3p. >120 cells/group were quantified in (E) and *n* = 4 biological replicates in (F). The dashed line in the violin plot shows medium value for the group, and the dotted lines represent two quartile lines in each group.
G, H    Mitochondrial membrane potential and ATP production in primary mouse ACMs isolated from WT and miR-574$^{-/-}$ mice under ISO treatment. Scale bar: 10 μm. >110 cells/group were quantified in bottom panel of (G) and *n* = 5 in (H).
I       Transfection of miR-574-5p/3p in the absence or presence of FAM210A overexpression. All the Western blot images were provided as representative data from 3 to 4 biologically replicated experiments. Quantitative data were shown in Fig EV4H.
J       Increased protein expression of FAM210A and ND1 in miR-574 KO hearts compared to WT hearts upon TAC surgery. *N* = 5 hearts (~110–150 CM cells) were measured per group. The dashed line in the violin plot shows medium value for the group and the dotted lines represent two quartile lines in each group.
K       The schematic model of the cardioprotective function and mechanism of miR-574-FAM210A axis against cardiac pathological remodeling.

Data information: Data were presented as mean ± SEM. *P* values were calculated by Kruskal–Wallis test with Dunn's multiple comparisons test (A, B, E), one-way ANOVA with Tukey's multiple comparisons test (F), two-way ANOVA with Tukey's multiple comparisons test (G, H), or one-way ANOVA with non-parametric Kruskal–Wallis test (J).
Source data are available online for this figure.

important roles in maintaining mitochondrial protein homeostasis and mitochondrial activity in other organs and related diseases. Our studies provide novel insights into the function and regulatory mechanisms of FAM210A in the modulation of MEG protein expression and mitochondrial activity. Although FAM210A is identified as the only common target gene of both miR-574-5p and miR-574-3p in our studies, other putative targets of miR-574-5p or miR-574-3p, for example, SDHA, HIBADH, LDHB, can exist and contribute to the pathophysiological effects (Appendix Fig S2D). We consider this is a limitation in our study and this open question warrants further investigation.

We propose that normalizing mitochondrial protein expression (e.g., removing an excessive amount of translation activity) could be applied to antagonize cardiac pathological remodeling in the treatment of heart disease. Our findings offer novel insights into the function of miR-574-5p/3p and their downstream effector FAM210A in regulation of mitochondrial protein synthesis and indicate that the miR-574 and miR-574-FAM210A axis may represent novel therapeutic targets for treatment of heart disease.

# Materials and Methods

### Human specimens

All human samples of frozen cardiac tissues (*n* = 23), including 17 samples from explanted failing hearts and 6 samples from donor non-failing hearts, were acquired from the Cleveland Clinic (Wu *et al*, 2020). This study was approved by the Material Transfer Agreement between the URMC and the Cleveland Clinic. Total RNA was extracted from human heart tissues using TRIzol reagent (Thermo Fisher Scientific) following instructions in the manual. For the mRNA detection, 1 μg of total RNA was used as a template for reverse transcription assay using the iScript cDNA Synthesis Kit (Bio-Rad); cDNA was used for detecting FAM210A and GAPDH expression. For the expression of miRNA, the total RNA (containing small RNAs) was subjected to miRNA reverse transcription using the miScript II RT Kit (Qiagen), following the manual. The RT

product was used for detection of miR-574-5p/3p and control RNA Snord68 using the miScript Primer Assay (Qiagen). Informed consent was obtained from all subjects and the experiments conformed to the principles set out in the WMA Declaration of Helsinki and the Department of Health and Human Services Belmont Report.

### Mice

Three Sanger MirKO ES cell lines for Mir574 (produced by Wellcome Trust Sanger Institute; Prosser *et al*, 2011) were purchased from the mutant mouse regional resource center (MMRRC). miR-574$^{-/-}$ global knockout chimera founder mice were generated in the Case Western Reserve University Transgenic Core Facility. We generated the miR-574 targeted male chimera mouse in the C57BL/6J background and performed germline transmission and backcrossed the mice to C57BL6J mice for more than 10 generations. A 182-bp DNA region was deleted, including the microRNA574 gene (96-bp for pre-miR-574 sequence) in the intron 1 of the host gene Fam114a1. Puromycin resistance cassette was removed by breeding with C57BL/6-Tg(Zp3-cre)93Knw/J mouse, which expresses Cre recombinase in oocytes, resulting in a null allele. For experiments with miR-574$^{-/-}$ mice, control mice of the same age (10–12 weeks) and gender (female mice for ISO model and male mice for TAC model) from littermates or sibling mating were used. All animal procedures were performed in accordance with the National Institutes of Health (NIH) and the University of Rochester Institutional guidelines. All the mice were maintained on a 12-hr light/dark cycle and fed with normal chow diet at the temperature of 22°C.

### Isoproterenol (ISO) injection and infusion model

Experimental mice are siblings generated from intercrosses of miR-574$^{+/-}$ mice. Age and background-matched WT and miR-574$^{-/-}$ female mice at 10–12 weeks of age were subjected to vehicle (saline) or ISO treatment. Procedure for ISO injection for the ISO treatment model (Fig 2): ISO or vehicle saline were administered to WT and miR-574$^{-/-}$ mice daily for 4 weeks using subcutaneous

injection (30 mg/Kg/day). Excised mouse hearts were flushed with saline to remove the blood, fixed in 10% formalin, and used for histological and immunoblotting analyses. Procedure for mini-osmotic pump implantation for the rescue model (Appendix Fig S5): Mouse was anesthetized using 2.0% isoflurane and placed on a heated surgical board. A side/upper back area skin incision was made, and a mini-osmotic pump was inserted subcutaneously set to deliver ISO or vehicle. The incision was then closed with 6-0 coated vicryl sutures in a subcuticular manner, and the animals were allowed to recover. The pumps were not removed and will remain for a time period of 4 weeks, after which the animals were euthanized. The sutures were removed after 2 weeks since the pumps were transplanted.

## Transverse aortic constriction (TAC) surgical model

Experimental mice were siblings generated from intercrosses of miR-574$^{+/-}$ mice. Age- and background-matched WT and miR-574$^{-/-}$ male mice at 10–12 weeks of age were subjected to Sham or TAC surgery. Each mouse was anesthetized using 2.0% isoflurane, placed on a surgical board with a heating pad (half inch plexiglass between the animal and the heating pad), and given buprenorphine SQ. A midline cervical incision was made to expose the trachea for visualizing oral intubation using a 22-gauge (PE90) plastic catheter. The catheter was connected to a volume-cycled ventilator supplying supplemental oxygen with a tidal volume of 225–250 μl and a respiratory rate of 120–130 strokes/min. Surgical plane anesthesia is subsequently maintained with 1–1.5% isoflurane. Procedure for left thoracotomy: Skin was incised, and the chest cavity was opened at the level of the 2$^{nd}$ intercostal space. Transverse section of the aorta was isolated. Transverse aortic constriction was created by placing a (6-0 silk) ligature securely around the trans-aorta and a 27-gauge needle, causing complete occlusion of the aorta. The needle was removed, restoring a lumen with severe stenosis. Lungs were reinflated and the chest was closed using Vicryl 6-0 suture. Muscle and skin were sutured using a Vicryl 6-0 suture in a running subcuticular pattern. Once the mouse was breathing on its own, it was removed from the ventilator and allowed to recover in a clean cage on a heated pad.

The mice were randomized for experiments using simple randomization with a specific ID number before animal procedures. All animal operations, including ISO infusion, TAC surgery, and echocardiography measurement, were performed by the Microsurgical Core surgeons. Sections and histology analysis were done by the Histology Core. The technicians from both Microsurgical Core and Histology Core were all blinded to the genotypes of the mice and tissue samples. For group size justification, we performed the power analysis using both G*power (Faul *et al*, 2007) version 3.1.9.6 and the function of power.anova.test in R version 3.5.3 (R Foundation for Statistical Computing, Vienna, Austria). The assumptions include the same standard variance in each study group, effect size = Difference of the means between study groups/common standard deviation, alpha level = 0.05, power = 0.9, and number of study groups. The effect size for specific experiments is assumed based on previous similar studies or literature. In previous experiences from our Microsurgical Core, we have observed a survival rate of ~ 90% after the TAC procedure. To offset the possible loss of mice per treatment, we added at least one mouse per treatment group.

## *In vivo* therapeutic model and rescue model using miRNA mimics

Based on recommendations from the nanoparticle user instruction, 75–100 μg miRNA with chemical modification (resistant to nuclease degradation *in vivo*) needs to be used per injection. In our experiments, miRNA mimics were used at the dose of 5 mg/Kg body weight in the volume of 150–200 μl for injections in male WT C57BL/6J mice (TAC model) or miR-574 KO mice (ISO model) (10–12 weeks old). The *mir*Vana® miRNA mimics for miR-574-5p/3p and scrambled *mir*Vana® miRNA mimics (negative control) (~100 μg) were dissolved in ~150–200 μl RNase-free water. The diluted miRNA mimics were incubated with 50 μl of nanoparticle-based *in vivo* transfection reagent (Altogen Biosystems, Cat. No. 5031) in sterile tubes for 20 mins at RT. Transfection enhancer (10 μl, Altogen Biosystems, Cat. No. 1799) was added to the mixture, vortexed gently, and incubated for 5 mins at RT. The nanoparticle-miRNA mimics complex was mixed with an appropriate volume of the sterile solution of 5% glucose (w/v), and delivered into the murine heart by intravenous tail vein injections (after mice are anesthetized using 2.0% isoflurane) once a week after TAC surgery following the manufacturer's instruction and a previous report (Wang *et al*, 2016).

In the rescue model, miR-574 KO mice (2 months old) were subcutaneously infused with Azlet osmotic minipumps filled with either saline or ISO (20 mg/Kg body weight) for 28 days. After 3 days of infusion, mice were injected with nanoparticle carrying miR-574-3p, miR-574-5p, both combined, or scrambled miRNA mimics (5 mg/Kg body weight) through the tail vein on day 4, 10, 16, and 22 using a 1 mm BD syringe. After the treatment period, mice were euthanized, and tissues were flash frozen and stored at −80°C until further analysis.

## Echocardiography

Echocardiographic image collection was performed using a Vevo2100 echocardiography machine (VisualSonics, Toronto, Canada) and a linear-array 40 MHz transducer (MS-550D). Image capture was performed in mice under general isoflurane anesthesia with heart rate maintained at 500–550 beats/min. LV systolic and diastolic measurements were captured in M-mode from the parasternal short axis. Fraction shortening (FS) was assessed as follows: % FS = (end diastolic diameter - end systolic diameter)/ (end diastolic diameter) x 100%. Left ventricular ejection fraction (EF) was measured and averaged in both the parasternal short axis (M-Mode) using the tracing of the end diastolic dimension (EDD) and end systolic dimension (ESD) in the parasternal long axis: % EF = (EDD-ESD)/EDD. Hearts were harvested at multiple endpoints depending on the study.

## Cell culture

AC16 adult human ventricle cardiomyocyte cells (SCC109, Sigma), primary mouse neonatal and adult cardiomyocytes, primary mouse adult cardiac fibroblasts, and HEK293T cells were used to address questions at the cellular and molecular level. Cells were cultured following the manual (Sigma, Cat. No. SCC109 for AC16). AC16 cells were cultured in DMEM/F12 containing 2 mM L-Glutamine, 12.5% FBS, and 1x Penicillin/Streptomycin Solution. For the

myocyte hypertrophy experiment, AC16 cells were treated with 10 μM ISO for 24 h. HEK293T cells were cultured in DMEM containing 2 mM L-Glutamine, 10% FBS, 1× Penicillin/Streptomycin Solution. Cell culture methods are described in the Appendix Materials and Methods for primary mouse neonatal and adult cardiomyocytes, and primary mouse adult cardiac fibroblasts. AC16 cells were purchased from Sigma and authenticated, and all the cells were tested for mycoplasma contamination using a detection kit.

## Luciferase activity assay

Reporter vectors were generated by inserting the Fam210a 3'UTR fragments containing wild-type or mutated seed sequence into the miRNA reporter vector pmirGlo (Promega). All the primers used to generate the reporter vector were listed in supplemental information. The wild-type 3'UTR fragment was amplified and ligated to pmirGlo. Then, 100 ng of wild-type or mutant reporter vectors were co-transfected with 20 nM miRNA (miR-574-5p or miR-574-3p) into HEK293T cells cultured in 24-well plate using lipofectamine 3000 (Thermo Fisher Scientific) following the manufacturer's instruction. Cells were collected 36 h after transfection. Additionally, firefly and renilla luciferase activity were detected by the Dual Luciferase Reporter Assay System (Promega).

## Immunofluorescence and confocal microscopy

Immunostaining of cells grown on coverglass or chambered slides: HEK293T, AC16 cells, primary neonatal or adult CMs were grown on the coverslips, and incubated with 100 nM MitoTracker Red CMXRos (Thermo Fisher Scientific) for 30 mins at 37°C before being fixed for 10 mins with 4% paraformaldehyde in PBS. Cells were washed with PBS for 3 × 5 mins and permeabilized using ice-cold 0.5% Triton X-100 in PBS for 5 mins. After blocking with 1% BSA in PBS, the coverslips were incubated with indicated primary antibodies (anti-FAM210A 1:1,000; anti-FLAG 1:2,000) in blocking solution (2% BSA in PBS) for 1 h at RT and then washed with PBS for 3x 5 mins. The coverslips were incubated with the Alex Fluor-488 conjugated secondary antibodies (Thermo Fisher Scientific, 1:1,000) in PBS and washed with PBS for 3 × 5 min. Coverslips were air-dried and placed on slides with antifade mounting medium (containing DAPI). The slides were imaged using an Olympus FV1000 confocal microscope.

## Wheat germ agglutinin (WGA) and phalloidin staining

WGA (5 mg) was dissolved in 5 ml of PBS (pH 7.4). We performed deparaffinization by following steps: (i) Xylene (100%) for 2 × 5 mins; (ii) Ethanol (100%) for 2 × 5 mins; (iii). Ethanol (95%) for 1 × 5 mins; (iv) ddH$_2$O for 2 × 5 min. The slides were kept in a pressure cooker for 10 min along with citrate buffer (10 mM, pH 6.0) for antigen retrieval. We quenched the slides with 0.1 M glycine in phosphate buffer (pH 7.4) for 1 h at RT. Circles were made with a Dako pen, and slides were blocked with goat normal serum for 30 min. 10 μg/ml of WGA-Alexa Fluor 488 (Sigma Aldrich) was applied to the slides for incubation for 1 h at RT. Slides were rinsed in PBST wash buffer 3 × 5 min followed by PBS for 5 min. A coverslip was placed on the slides with an antifade solution (containing DAPI) for imaging. Five different cross-sectional

areas were selected, and the cell size of at least 500 CM cells were measured per area. For primary murine CM cell culture, Alexa Fluor™ 594 Phalloidin (Thermo Fisher Scientific, Cat. No. A12381) was used to measure the cell size following the instruction from the manual. Primary CM cells were treated with 10 μM ISO for 24 h for measuring cell size using Phalloidin. Additionally, primary CM cells were treated with 10 μM ISO for 48 h for the Trypan blue staining and TUNEL assay. Cultured CMs were fixed using a 4% paraformaldehyde in PBS for 10 min, washed with PBS, and permeabilized using 0.2% Triton X-100 for 10 min. Cells were blocked in 2% BSA/PBS for 1 h and stained with Alexa Fluor™ 594 Phalloidin in 1:1,000 dilution for 30 min at RT. The stained cells were gently washed with PBS for 3 × 5 min, and the slides were mounted using a mounting medium with DAPI.

## Picrosirius red staining

Paraffin-embedded tissue sections were deparaffinized and incubated in a picrosirius red solution (Abcam, Cat. No. ab150681) at RT for 1 h. Then, slides were subjected to 2 washes of 1% acetic acid and 100% of ethyl alcohol and then mounted in a resinous medium. Images were captured using the PrimeHisto XE Histology Slide Scanner (Carolina). Six images were selected from each group for analysis. Total collagen content was determined for the whole heart images using an Image J software.

## Terminal deoxynucleotidyl transferase dUTP nick end labeling (TUNEL) staining

The tissue sections were washed with PBS twice and fixed using 4% paraformaldehyde for 20 min. Cells were permeabilized with 0.5% of Triton X-100 for 5 min, and incubated in TUNEL reaction mixture (In Situ Cell Death Detection Kit; Sigma, 11684795910) for 1 h at 37°C in the dark. Finally, cells were washed with PBS for 3 × 5 min, air-dried, and mounted with DAPI-containing antifade medium. Images were captured using a BX51 microscope (Olympus).

## Determination of cellular ATP content

The cellular ATP content was determined using a Molecular Probes ATP Determination Kit (Thermo Fisher Scientific, A22066) according to the manufacturer's instructions. Briefly, cells (or heart lysates) were incubated in 1% (w/v) of Trichloroacetic acid and centrifuged at 9,391 *g* for 5 min at 4°C. The supernatant was collected and neutralized by Tris–HCl (0.1 M, pH 9.0), then mixed with the bioluminescent reagent. After incubation, bioluminescent signals were read using the HTX microplate reader (BioTek Instruments). ATP concentrations were determined in all samples based on a standard curve and normalized by the total protein mass.

## Measurement of mitochondrial membrane potential

The mitochondrial membrane potential was measured using TMRE (Tetramethylrhodamine ethyl ester, Thermo Fisher Scientific, Cat. No. T669) according to the manufacturer's protocol. Isolated adult mouse CMs were loaded with TMRE at 25 nM concentration for 30 mins at RT. After incubation, cells were washed with 1x PBS for 3

**The paper explained**

**Problem**

The expression and stoichiometric balance of nuclear-encoded mitochondrial genes (NEMG) and mitochondrial-encoded genes (MEG)-derived proteins are tightly regulated to maintain mitochondrial homeostasis and normal cardiac function. Aberrant expression of mitochondrial proteins impairs cardiac function and causes heart disease. The molecular mechanism of mitochondrial dysfunction in heart disease caused by unbalanced mitochondrial protein expression and the potential therapeutic approaches for treatment remains largely unknown.

**Results**

Here, we discover guide and passenger strands of microRNA-574, miR-574-5p, and miR-574-3p, as new cardioprotective factors in the heart. We demonstrate that manipulation of miR-574 level by either genetic knockout or exogenous injection influences the disease phenotype in both isoproterenol and transverse aortic constriction induced heart failure models. Further, we demonstrate that FAM210A, a common downstream target of miR-574-5p and miR-574-3p, serves as a novel regulator of mitochondrial-encoded protein expression. The miR-574-FAM210A axis functions to modulate mitochondrial-encoded protein expression and influences pathological cardiac remodeling.

**Impact**

Our findings indicate that the miR-574 and miR-574-FAM210A axis may represent novel therapeutic targets for cardiac pathological remodeling and heart failure, for which a very limited number of effective therapies are available to date. Normalizing mitochondrial protein expression may serve as a potential therapeutic strategy for treatment of heart disease.

times and submerged in live cell imaging solution. Fluorescence images were captured using a laser scanning confocal microscope (Olympus).

**Dihydroethidium (DHE) staining**

DHE staining was performed in both isolated live murine cardiomyocyte cells and frozen heart tissue sections for detection of intracellular reactive oxygen species (ROS). Primary adult CMs were isolated from WT and miR-574$^{-/-}$ mice and cultured using the standard procedure. Also, heart sections were prepared by the frozen section procedure from WT and miR-574$^{-/-}$ mice. In situ superoxide radical's production in live CMs and frozen sections was measured with an oxidative fluorescent dye called Dihydroethidium (DHE, Thermo Fisher Scientific). Live CM cells were plated and then treated with vehicle (0.5% DMSO) and ISO for 24 h. The cell or heart tissue sections were incubated with 10 μM DHE for 20 mins at 37°C in the dark and washed with PBS for 3x 15 mins. Sections were air-dried and mounted with a coverslip. Images were captured using an Upright digital immunofluorescence microscope (BX51, Olympus). Four images were taken from each section. Fluorescence signal intensities were quantified with Image J software.

**Statistical analysis**

All quantitative data were presented as mean ± SEM, and analyzed using Prism 8.3.0 software (GraphPad). We used a Kolmogorov–Smirnov test to assess if the data was normally distributed. For a comparison between 2 groups, an unpaired Mann–Whitney test for not normally distributed data and an unpaired two-tailed Student t test for normally distributed data were performed. For multiple comparisons among ≥ 3 groups, a one-way or two-way ANOVA with Tukey's method for post hoc comparisons and non-parametric Kruskal–Wallis test with Dunn's multiple comparisons were performed. Two-sided $P$ values < 0.05 were considered to indicate statistical significance. Specific statistical methods and post hoc tests were described in the figure legends.

# Data availability

RNA-Seq data produced and used in this study were deposited on under accession number GSE149168 (http://www.ncbi.nlm.nih.gov/geo/query/acc.cgi?acc = GSE149168) in Gene Expression Omnibus (GEO) database.

**Expanded View** for this article is available online.

# Acknowledgments

We are grateful to Yonggang Zheng, Jordan Pappas, Virginia Aswad, and Matthew Auguste for critical reading of the manuscript and Qiuqing Wang for statistical consulting. We appreciate the technical assistance from Ronald Conlon (Case Western Reserve University), Orazio Slivano, and Deanne Mickelsen (Aab CVRI) in generating miR-574$^{-/-}$ mice, histology, and surgical operations, respectively. RNA sequencing was performed by the Genomics Core at the Case Western Reserve University, and the primary data analysis was done by Jason R Myers from the Genomics Research Center at the University of Rochester. None of the authors have any financial conflict of interest related to the research described in this manuscript. This work was supported in part by National Institutes of Health (R56 HL132899-01, R01 HL132899, and R01HL147954 to P.Y.; R01 HL134910 and HL088400 to C.Y.; R00 EY025290 and R01 GM127652 to T.Y.), University of Rochester CTSA award number (UL1 TR002001 to P.Y.) from the National Center for Advancing Translational Sciences of the National Institutes of Health (content is solely the responsibility of the authors and does not necessarily represent the official views of the National Institutes of Health), Scientist Development Grant (13SDG15970006 to P.Y.) and postdoctoral fellowship (19POST34400013 to J.W.) from the American Heart Association, and start-up funds from Aab Cardiovascular Research Institute of University of Rochester Medical Center (to P.Y.). Funding for open access charge: National Institutes of Health.

## Author contributions

PY conceived the study, provided the funding, and wrote the manuscript. PY, JW, and KCVS designed the experiments and analyzed the data. PY, JW, KCVS, FJ, and OH carried out the experimental work. AM performed mouse TAC surgery. KW and SG contributed to the mass spectrometry analysis. WHWT provided critical human samples for our studies. TY, ES, and CY provided expertise, technical assistance, and feedback. All the authors discussed the results and had the opportunity to comment on the manuscript.

## Conflict of interest

The authors declare that they have no conflict of interest.

## For more information

i  The GTEx Portal database: http://gtexportal.org/home/.

ii  Gene Expression Omnibus (GEO) database: https://www.ncbi.nlm.nih.gov/geo/.

iv  IMPC database: https://www.mousephenotype.org.

iii  UCSC Genome Browser: https://genome.ucsc.edu.

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
