## [Review Process File · EMBO Molecular Medicine]

MicroRNA-574 Regulates FAM210A Expression and Influences Pathological Cardiac Remodeling

Jiangbin Wu, Kadiam C Venkata Subbaiah, Feng Jiang, Omar Hadaya, Amy Mohan, Tingting Yang, Kevin Welle, Sina Ghaemmaghami, Wai Hong Wilson Tang, Eric Small, Chen Yan, Peng Yao

DOI: [10.15252/emmm.202012710](https://doi.org/10.15252/emmm.202012710)

Corresponding authors: Peng Yao (Peng_Yao@urmc.rochester.edu)

Review Timeline:

Submission Date:	24th May 20
Editorial Decision:	16th Jun 20
Revision Received:	23rd Sep 20
Editorial Decision:	16th Oct 20
Revision Received:	17th Nov 20
Accepted:	19th Nov 20

Editor: Lise Roth

Transaction Report:

16th Jun 2020

Dear Prof. Yao,

Thank you for submitting your work to EMBO Molecular Medicine. We have now heard back from the three referees who agreed to evaluate your manuscript. As you will see below, the reviewers raise substantial concerns on your work, which preclude its publication in EMBO Molecular Medicine in its current form.

The reviewers find that the question addressed by the study is of potential interest, and appreciate the novelty of the findings. However, they remain unconvinced that some of the major conclusions are sufficiently supported by the data. They thus raise the following major issues:

- the global approaches used in vivo (global KO and iv injection of miRNA mimetics) do not allow for analysis of the role of miR574 on the different cell types. While generating cell-specific knock-out mice would not be realistic in a reasonable time frame, in vitro experiments should be performed to address this issue.
- the effects on mitochondria are not adequately characterized.

If you feel you can satisfactorily address these points as well as the other points listed by the referees, you may wish to submit a revised version of your manuscript.

Addressing the reviewers' concerns in full will be necessary for further considering the manuscript in our journal, and acceptance of the manuscript will entail a second round of review. EMBO Molecular Medicine encourages a single round of revision only and therefore, acceptance or rejection of the manuscript will depend on the completeness of your responses included in the next, final version of the manuscript. For this reason, and to save you from any frustrations in the end, I would strongly advise against returning an incomplete revision.

To submit your manuscript, please follow this link:

Link Not Available

When submitting your revised manuscript, please carefully review the instructions that follow below. Failure to include requested items will delay the evaluation of your revision:

- 1) A .docx formatted version of the manuscript text (including legends for main figures, EV figures and tables). Please make sure that the changes are highlighted to be clearly visible.
- 2) Individual production quality figure files as .eps, .tif, .jpg (one file per figure).
- 3) A .docx formatted letter INCLUDING the reviewers' reports and your detailed point-by-point responses to their comments. As part of the EMBO Press transparent editorial process, the point-

by-point response is part of the Review Process File (RPF), which will be published alongside your paper.

4) A complete author checklist, which you can download from our author guidelines (<https://www.embopress.org/page/journal/17574684/authorguide#submissionofrevisions>). Please insert information in the checklist that is also reflected in the manuscript. The completed author checklist will also be part of the RPF.

5) Before submitting your revision, primary datasets produced in this study need to be deposited in an appropriate public database (see <https://www.embopress.org/page/journal/17574684/authorguide#dataavailability>). Please remember to provide a reviewer password if the datasets are not yet public. The accession numbers and database should be listed in a formal "Data Availability" section (placed after Materials & Method). Please note that the Data Availability Section is restricted to new primary data that are part of this study.

6) We would also encourage you to include the source data for figure panels that show essential data. Numerical data should be provided as individual .xls or .csv files (including a tab describing the data). For blots or microscopy, uncropped images should be submitted (using a zip archive if multiple images need to be supplied for one panel). Additional information on source data and instruction on how to label the files are available at .

7) Our journal encourages inclusion of *data citations in the reference list* to directly cite datasets that were re-used and obtained from public databases. Data citations in the article text are distinct from normal bibliographical citations and should directly link to the database records from which the data can be accessed. In the main text, data citations are formatted as follows: "Data ref: Smith et al, 2001" or "Data ref: NCBI Sequence Read Archive PRJNA342805, 2017". In the Reference list, data citations must be labeled with "[DATASET]". A data reference must provide the database name, accession number/identifiers and a resolvable link to the landing page from which the data can be accessed at the end of the reference. Further instructions are available at .

8) We replaced Supplementary Information with Expanded View (EV) Figures and Tables that are collapsible/expandable online. A maximum of 5 EV Figures can be typeset. EV Figures should be cited as 'Figure EV1, Figure EV2" etc... in the text and their respective legends should be included in the main text after the legends of regular figures.

- Additional Tables/Datasets should be labeled and referred to as Table EV1, Dataset EV1, etc. Legends have to be provided in a separate tab in case of .xls files. Alternatively, the legend can be supplied as a separate text file (README) and zipped together with the Table/Dataset file. See detailed instructions here:

9) The paper explained: EMBO Molecular Medicine articles are accompanied by a summary of the articles to emphasize the major findings in the paper and their medical implications for the non-specialist reader. Please provide a draft summary of your article highlighting

10) For more information: There is space at the end of each article to list relevant web links for further consultation by our readers. Could you identify some relevant ones and provide such information as well? Some examples are patient associations, relevant databases, OMIM/proteins/genes links, author's websites, etc...

11) Author contributions: the contribution of every author must be detailed in a separate section (before the acknowledgments).

12) A Conflict of Interest statement should be provided in the main text

13) Every published paper now includes a 'Synopsis' to further enhance discoverability. Synopses are displayed on the journal webpage and are freely accessible to all readers. They include a short stand first (maximum of 300 characters, including space) as well as 2-5 one-sentences bullet points that summarizes the paper. Please write the bullet points to summarize the key NEW findings. They should be designed to be complementary to the abstract - i.e. not repeat the same text. We encourage inclusion of key acronyms and quantitative information (maximum of 30 words / bullet point). Please use the passive voice. Please attach these in a separate file or send them by email, we will incorporate them accordingly.

Please also suggest a striking image or visual abstract to illustrate your article. If you do please provide a png file 550 px-wide x 400-px high.

14) As part of the EMBO Publications transparent editorial process initiative (see our Editorial at <http://embomolmed.embopress.org/content/2/9/329>), EMBO Molecular Medicine will publish online a Review Process File (RPF) to accompany accepted manuscripts.

In the event of acceptance, this file will be published in conjunction with your paper and will include the anonymous referee reports, your point-by-point response and all pertinent correspondence relating to the manuscript. Let us know whether you agree with the publication of the RPF and as here, if you want to remove or not any figures from it prior to publication.

I look forward to receiving your revised manuscript.

Yours sincerely,

Lise Roth

Lise Roth, PhD
Editor
EMBO Molecular Medicine

Photos 400-800 DPI

*Additional important information regarding figures and illustrations can be found at <http://bit.ly/EMBOPressFigurePreparationGuideline>

***** Reviewer's comments *****

Referee #1 (Comments on Novelty/Model System for Author):

Only suggestion is that a tissue specific KO of the mouse model tested should have been generated to provide more detailed cell-specific information about the loss of this particular miR.

Referee #1 (Remarks for Author):

The manuscript entitled, "MicroRNA-574 regulates FAM210A expression and influences pathological cardiac remodeling" by Wu et. al. sought to determine whether pathogenic cardiac stressors can induce expression of miR-574 to modulate FAM210A. They show that the increase in miR-574 and subsequent increase in FAM210A, expression modulates the expression of mitochondrial encoded proteins to elicit a protective response that antagonizes pathological cardiac remodeling. This is a nicely written manuscript with extensive high quality data to support this premise. The only major comment is with regards to the choice of animal models made; it seems it would have been more effective to generate tissue specific microRNA-574 KO mice, as opposed to whole body KO mice where it remains unclear which tissue types contribute to which physiological effects in the heart. Nonetheless, the data are strong, with significant results that provide substantial novelty.

Some minor additional suggested edits:

- 1) Echo images throughout the paper are not convincing. Please replace with images that better represent the data in the tables. In addition, several cardiac functional parameters are missing from the echo tables, including LVIDd and LVIDs, which assesses the effects of the KO on dilatation, particularly given the changes in LVPW are minimal.
- 2) The apoptosis TUNEL experiments throughout the paper are also not convincing, particularly since only 1-2 cells are indicated in each panel and it is not clear from the image if the apoptotic cell is even a cardiomyocyte.
- 3) Authors suggest that the KO reveals a "protective role of miR-574-5p and miR-574-3p." However, it is not clear that the KO mouse reveals this; rather, it reveals that one OR both may be involved in the protective effects.
- 4) In this regard, the data in Fig 4 are difficult to interpret. To me, it seems that the addition of the miR-574-3p mimetic induces greater hypertrophy, as demonstrated by the H&E image in 4B and the echo in 4D, as compared to WT. It is clear this miR protects from fibrosis, however. In contrast, it seems the miR-574-5p mimetic better normalizes the heart, but has more minimal effects on fibrosis. How do the authors reconcile this?
- 5) How did the authors conclude that both miRs are needed for cardioprotection and why did they only hone in on the one effector that binds to both miR-574-5p and miR-574-3p for effects on cardioprotection? It seems likely there are individual targets that could play a differential role in this process (ie, in hypertrophy vs fibrosis; see my comment about fig 4). It seems the authors may be missing critical functional mechanisms by focusing in on only this one target. At the very least, this should be discussed as a limitation in the discussion section of the paper.
- 6) Authors should also directly test the effects of the mimetics in vivo in their KO mice to see if they can rescue the deleterious effects of pathological hypertrophy in response to TAC.

Referee #2 (Remarks for Author):

This study by Wu and colleagues assess the contribution of microRNA-574 to cardiac stress responses and pathophysiology. The work revealed FAM210A as important target for miR574 in the heart and a mechanism by which FAM210A regulates mitochondrial translation through an interaction with the elongation factor EF-Tu. Overall this study contains high quality and interesting data that are informative and novel. The use of loss and gain of function for miR574, as well as more than one cardiac stress model, provides a comprehensive assessment of the contribution of miR574 in the heart. However, the below concerns should be addressed to corroborate some of their findings:

1. It would be important to test the cardiomyocyte specificity of miR574 induction post stress using primary cardiac cells. This is particularly critical considering the use of global strategies for in vivo manipulation of the levels of this microRNA.
2. Analysis of miR574 induction in the stressed heart (perhaps following a time course of pressure overload) would also represent important information. This will help interpret the somewhat modest cardioprotective effect of mimetic oligomers and understand if starting administering them earlier than 7 days post pressure overload could have provided a stronger result.
3. An important missing control when addressing the mechanistic part of the study is measurements of the number of mitochondria in all mouse mutants and treatments utilized in this

study. Assessing mitochondria DNA content in relation to nuclear DNA content is critical to assess if the miR574-FAM210A axis indeed regulates mitochondrial protein translation instead of contributing to regulation of mitochondrial content in cardiomyocytes. I also suggest performing electron microscopy to assess mitochondria morphology in the heart of miR574 KO mice.

4. It would be nice to include a northern blot (similar to what you have in supplemental figure 1A) to corroborate organ enrichment of Fam210A expression for figure 6, and correlate this data to miR574 expression pattern.

Minor points

1. In figure 2D and 3D, when showing the zoomed picosirius red images, please pick a similar location from each heart.

2. In figure 4, the overall effect of mimetic oligomers in rescuing cardiac function is not very robust. Please replace Echo images shown in figure 4D to better represent the result. Accordingly, please also moderate the statements related to the rescue potential of the used treatment.

3. Also, apoptosis is not a major mechanism for cardiomyocyte cell death during pressure overload. It is likely that the few TUNEL positive nuclei observed in the myocardium might actually originate from non-myocyte populations. Perhaps including that assay as supplemental data would be better suited.

Referee #3 (Comments on Novelty/Model System for Author):

While the cardiac function of miR-574 is unknown and the authors make use of a new knockout mouse model, the cellular function of miR-574 remains unclear and the in vivo relevance of the proposed mechanism is not investigated.

Since a global knockout is used and the mimics are also delivered to all cardiac cell types the authors would have to study the function of miR-574 in multiple cell types to study its true contribution to cardiac function and remodeling in vivo.

Referee #3 (Remarks for Author):

In this manuscript the authors show that miR-574 is induced in the heart during cardiac disease in both mice and human. Global genetic deletion of miR-574 does not induce an overt phenotype under baseline conditions but exacerbates remodeling during cardiac stress. Conversely, intravenous delivery of miR-574 mimic inhibits TAC induced pathological remodeling. The authors next define FAM210A as a direct target of both strands of miR-574 and show in vitro that this regulates cellular mitochondrial function.

MicroRNAs have been shown to be important players for cardiac biology and disease including mitochondrial function. While the observed in vivo effects are striking, some key data points are lacking to provide sufficient comfort regarding the proposed mechanism.

Some key issues:

- Which cardiac cells express miR-574 and how does this change during disease? Is this regulation different between mice and human?

- Which cardiac cell type is targeted by the miR-574 mimics and how does this effect the different cell types?
- Which tissues besides the heart is the miR mimic delivered to and how does this influence the observed phenotype?
- The authors show that FAM210A is a direct target of both strands of miR-574. The authors are also able to show a regulation of FAM210A in the knockout mice. However, it remains unclear how this effects downstream regulators of mitochondrial function in vivo or how this effects mitochondrial function.

Editor's summarized major points:

The reviewers find that the question addressed by the study is of potential interest, and appreciate the novelty of the findings. However, they remain unconvinced that some of the major conclusions are sufficiently supported by the data. They thus raise the following major issues:

- the global approaches used in vivo (global KO and iv injection of miRNA mimetics) do not allow for analysis of the role of miR574 on the different cell types. While generating cell-specific knock-out mice would not be realistic in a reasonable time frame, in vitro experiments should be performed to address this issue.

Response: *We have performed multiple in vitro experiments (including newly added and previously shown experiments) to address the miR-574 expression pattern and cell type specific cardioprotective effects of miR-574-5p or miR-574-3p, which are presented in Figures EV1B, EV1C, EV1F, EV3G, EV3H, EV3I, 7F-G, 8A-I, S4A-B, EV4A-E, and EV4H-I.*

- the effects on mitochondria are not adequately characterized.

Response: *We have provided more adequate characterizations of the effects of miR-574 loss of function on mitochondria in addition to the data shown in the last version, which are all presented in Figures 8G-H, 8J, EV4F, EV4G, EV5A, EV5B, EV5C, EV5D, and EV3E.*

Referee #1 (Comments on Novelty/Model System for Author):

Only suggestion is that a tissue specific KO of the mouse model tested should have been generated to provide more detailed cell-specific information about the loss of this particular miR.

Referee #1 (Remarks for Author):

The manuscript entitled, "MicroRNA-574 regulates FAM210A expression and influences pathological cardiac remodeling" by Wu et. al. sought to determine whether pathogenic cardiac stressors can induce expression of miR-574 to modulate FAM210A. They show that the increase in miR-574 and subsequent increase in FAM210A, expression modulates the expression of mitochondrial encoded proteins to elicit a protective response that antagonizes pathological cardiac remodeling. This is a nicely written manuscript with extensive high quality data to support this premise. The only major comment is with regards to the choice of animal models made; it seems it would have been more effective to generate tissue specific microRNA-574 KO mice, as opposed to whole body KO mice where it remains unclear which tissue types contribute to which physiological effects in the heart. Nonetheless, the data are strong, with significant results that provide substantial novelty.

Response: *We appreciate this constructive suggestion from the reviewer. As advised by the editor, generating cell-specific knockout mice would be challenging in a short term. It is our future research goal to generate a cell type specific miR-574 knockout mouse model to further investigate the function of miR-574 in different cell types in the heart.*

To address the reviewer's question using the available miR-574 KO mouse model, we measured miR-574-5p and miR-574-3p in isolated primary cardiomyocytes and cardiac fibroblasts from hearts under TAC surgery for 3 days. We found that miR-574-5p and miR-574-3p were significantly induced in CMs and CFs, respectively and the other strand was only slightly induced in each cell type (New Figure EV1C). We have also performed in vitro experiments using the anti-miRNA inhibitors (loss-of-function) and miRNA mimics (gain-of-function) transfection in human AC16 CM cells and confirmed the anti-hypertrophic (Figure EV4C, D) and mitochondrial protective (Figure 8E, F) effects of miR-574-5p/3p. The anti-hypertrophic effect of miR-574-5p/3p mimics was also confirmed in primary mouse neonatal CMs (New Figure EV3G). More importantly, the anti-hypertrophic, anti-apoptotic, and

mitochondrial protective activities of miR-574 were validated in isolated primary mouse adult CM cells from WT and miR-574 KO hearts (Figure EV2A, B and Figure 8G, H). In contrast, we did not observe obvious effects of both miR-574-5p and miR-574-3p on the proliferation of primary mouse CFs (new Figure EV3H). miR-574-3p (not miR-574-5p) reduced α -SMA protein expression in TGF- β -activated primary mouse CFs though both miRNA strands inhibited α -SMA protein expression at baseline (new Figure EV3I). These results suggest that exogenous miR-574-5p and miR-574-3p mimics may primarily target CMs as a synergistic pair and could also target CFs as a complementary mechanism. Please see Page 11 and 14 in the text. Vertical lines were drawn in the left margin area to highlight the major changes.

Some minor additional suggested edits:

1) Echo images throughout the paper are not convincing. Please replace with images that better represent the data in the tables. In addition, several cardiac functional parameters are missing from the echo tables, including LVIDd and LVIDs, which assesses the effects of the KO on dilatation, particularly given the changes in LVPW are minimal.

Response: *According to the reviewer's question, we have replaced some images in Figure 3F and Figure 4D to better represent the data in the tables. We have also included LVIDd and LVIDs as well as cardiac output, stroke volume, and LV mass in the supplemental Tables.*

2) The apoptosis TUNEL experiments throughout the paper are also not convincing, particularly since only 1-2 cells are indicated in each panel and it is not clear from the image if the apoptotic cell is even a cardiomyocyte.

Response: *We agree with the reviewer that CM apoptosis is not a strong phenotype during pressure overload and TUNEL signal may not come from CM cells. As also suggested by reviewer 2, we have moved TUNEL experimental data to supplemental figures (Figure EV2E, EV3F) and moderated the statement in the text (...cardiac apoptosis (though occurring at modest level under pressure overload)...). In Figure EV2B, we have shown that cultured primary mouse adult CMs from miR-574 KO mice are more susceptible to ISO-induced apoptosis. We believe that this experiment confirmed the autonomous effects of loss-of-function of miR-574 on CM apoptosis in the cell culture system in vitro using isolated primary CM cells.*

3) Authors suggest that the KO reveals a "protective role of miR-574-5p and miR-574-3p." However, it is not clear that the KO mouse reveals this; rather, it reveals that one OR both may be involved in the protective effects.

Response: *We have modified the statement as suggested by the reviewer. We changed "protective role of miR-574-5p and miR-574-3p" to "potential protective effect of either one or both of miR-574-5p and miR-574-3p strands".*

4) In this regard, the data in Fig 4 are difficult to interpret. To me, it seems that the addition of the miR-574-3p mimetic induces greater hypertrophy, as demonstrated by the H&E image in 4B and the echo in 4D, as compared to WT. It is clear this miR protects from fibrosis, however. In contrast, it seems the miR-574-5p mimetic better normalizes the heart, but has more minimal effects on fibrosis. How do the authors reconcile this?

Response: *We greatly appreciate the reviewer's thoughtful comment on Figure 4. The representative images for miR-574-3p mimics panel in Figure 4B and 4D did not present well the quantitative results shown in the right panel of WGA staining results. We replaced the images as also suggested by the reviewer for question 1. Based on the quantitative result in WGA staining in Figure 4B, miR-574-3p mimics reduced cardiac hypertrophy to a similar extent than miR-574-5p mimics upon TAC surgical stress (but definitely not inducing hypertrophy by miR-574-3p). We agree with the reviewer that miR-574-5p mimics show slightly better inhibition of cardiac hypertrophy and less inhibition of fibrosis than miR-574-3p mimics. We think that*

there might be two possible explanations: 1) The loading efficiency of miR-574-5p and miR-574-3p onto RNA-induced silencing complex (RISC) with Ago2 may be different after miRNAs entering CM and CF cells via nanoparticle delivery; 2) As the reviewer commented in question 5, the function and contribution of other putative individual targets of miR-574-5p/3p cannot be ruled out to play a differential role in cardiac remodeling (e.g., in hypertrophy versus fibrosis) and warrant further investigation. Consistent with our *in vivo* findings, in our new Figure EV3G-I, we found that miR-574-5p showed stronger anti-hypertrophic activity but weaker inhibition of myofibroblast activation compared to miR-574-3p. We have added these points in the discussion. Please see Page 20 in the text.

5) How did the authors conclude that both miRs are needed for cardioprotection and why did they only done in on the one effector that binds to both miR-574-5p and miR-574-3p for effects on cardioprotection? It seems likely there are individual targets that could play a differential role in this process (ie, in hypertrophy vs fibrosis; see my comment about fig 4). It seems the authors may be missing critical functional mechanisms by focusing in on only this one target. At the very least, this should be discussed as a limitation in the discussion section of the paper.

Response: We highly appreciate the constructive suggestions from the reviewer. We agree with the reviewer that we may not conclude that both miRNAs are essential for cardioprotection. Instead, we have modified the statement from “protective role of miR-574-5p and miR-574-3p” to “potential protective effects of either one or both of miR-574-5p and miR-574-3p strands” on Page 12. We identified bona fide miR-574-5p/3p targets by performing RNA-Seq of the total RNA from the hearts of WT and miR-574^{-/-} mice at baseline. We found that 34 genes are significantly upregulated at the mRNA level. Among these, 8 genes contain the putative miR-574-5p target seed sequence, while one single gene contains the putative miR-574-3p target seed sequence. Although FAM210A is identified as the only common target gene of both miR-574-5p and miR-574-3p in our studies, the function and contribution of other putative individual targets of miR-574-5p/3p cannot be ruled out to play a differential role in cardiac remodeling (e.g., in hypertrophy versus fibrosis) and warrant further investigation. We added Appendix Figure S2D to show several other potential regulated target transcripts by miR-574-5p in addition to Fam210a. As suggested by the reviewer, we have stated this as a limitation to our study in the discussion. Please see Page 22 in the text.

6) Authors should also directly test the effects of the mimetics *in vivo* in their KO mice to see if they can rescue the deleterious effects of pathological hypertrophy in response to TAC.

Response: We appreciate this important suggestion from the reviewer. The more pronounced cardiac remodeling could be due to *in-cis* effects from the DNA deletion in the genomic locus of miR-574. To confirm the *in-trans* effects of miR-574 in cardioprotection, we have already tested the effects of the miR-574 mimetics *in vivo* in our miR-574 KO mice to see if they could rescue the deleterious effects of pathological hypertrophy in response to ISO stimulation (we did not include the data in our original submission, but we included it in this revised version). As we showed in new Appendix Figure S5, we injected miR-574-5p/3p mimics into miR-574^{-/-} mice and could significantly rescue the cardiac disease phenotype under ISO treatment including reducing CM hypertrophy and cardiac fibrosis (accompanied by reduced hypertrophy and fibrosis marker gene expression), and myocyte death. We hope that these data will be able to address the similar question from reviewer regarding TAC model because we used both ISO and TAC models in which cardiac hypertrophy and fibrosis are activated as common pathological remodeling process. Please see Page 20 in the text.

Referee #2 (Remarks for Author):

This study by Wu and colleagues assess the contribution of microRNA-574 to cardiac stress

responses and pathophysiology. The work revealed FAM210A as important target for miR574 in the heart and a mechanism by which FAM210A regulates mitochondrial translation through an interaction with the elongation factor EF-Tu. Overall this study contains high quality and interesting data that are informative and novel. The use of loss and gain of function for miR574, as well as more than one cardiac stress model, provides a comprehensive assessment of the contribution of miR574 in the heart. However, the below concerns should be addressed to corroborate some of their findings:

1. It would be important to test the cardiomyocyte specificity of miR574 induction post stress using primary cardiac cells. This is particularly critical considering the use of global strategies for in vivo manipulation of the levels of this microRNA.

Response: *We appreciate this constructive suggestion from the reviewer. We isolated cardiac myocytes and fibroblasts in 3-day post TAC hearts from WT mice and extracted miR-574-5p and miR-574-3p. We found that miR-574-5p and miR-574-3p were significantly induced in CMs and CFs, respectively and the other strand was only slightly induced in each cell type (new Figure EV1C). These observations are consistent with increased miR-574-5p and miR-574-3p observed in cultured adult mice CM with ISO treatment (compared to vehicle treatment) and therefore indicate the higher miR-574-5p induction in CMs than in CFs. Please see Page 11 in the text. Vertical lines were drawn in the left margin area to highlight the major changes.*

2. Analysis of miR574 induction in the stressed heart (perhaps following a time course of pressure overload) would also represent important information. This will help interpret the somewhat modest cardioprotective effect of mimetic oligomers and understand if starting administering them earlier than 7 days post pressure overload could have provided a stronger result.

Response: *In the new Figure EV1D, E, we measured miR-574-5p and miR-574-3p expression level in the stressed hearts at different time points post-TAC surgery, including 3-day, 7-day, 14-day and 21-day. We observed that miR-574-5p and miR-574-3p were induced by ~2-3 folds on day 3 through day 21 (4 week time point was shown in Figure 1E). We agree with the reviewer that this may explain that modest cardioprotective effects of injected mimetic oligomers starting from day 7 and that starting to administer the oligomers earlier than 7 days might provide a stronger cardioprotective effect. We chose to inject miR-574 mimics starting on day 7 based on a scenario of a therapeutic-like model rather than a preventive model (start injection at earlier time point such as 1-3 days post TAC). Please see Page 11 in the text.*

3. An important missing control when addressing the mechanistic part of the study is measurements of the number of mitochondria in all mouse mutants and treatments utilized in this study. Assessing mitochondria DNA content in relation to nuclear DNA content is critical to assess if the miR574-FAM210A axis indeed regulates mitochondrial protein translation instead of contributing to regulation of mitochondrial content in cardiomyocytes. I also suggest performing electron microscopy to assess mitochondria morphology in the heart of miR574 KO mice.

Response: *According to the reviewer's question, we measured the copy number of mitochondrial DNA in relation to nuclear DNA content. We did not observe obvious changes in mitochondrial DNA content in miR-574 KO hearts compared to WT hearts at baseline or 3 days post TAC surgery (new Figure EV5C). We also did not observe changes in mitochondrial DNA content for miR-574-5p/3p mimics transfection in AC16 human CM cells (new Figure EV4I). We performed electron microscopy to assess mitochondrial morphology in the heart of miR-574 KO vs. WT mice at baseline and under ISO stimulation. We found that the morphology does not change much in miR-574 KO heart at baseline but mitochondria underwent more swelling, reduced cristae formation, and overall disarray upon the ISO stress than WT hearts (see new*

Figure EV4F, G), suggesting miR-574 KO renders the heart prone to more severe mitochondrial dysfunction under cardiac stresses. Please see Page 18 in the text.

4. It would be nice to include a northern blot (similar to what you have in supplemental figure 1A) to corroborate organ enrichment of Fam210A expression for figure 6, and correlate this data to miR574 expression pattern.

Response: *The northern blot data was obtained in Cleveland Clinic before I established my own lab. I have not obtained a license to use radioactivity in my current position in Aab Cardiovascular Research Institute. It will take several months before we can get permission to do Northern blot. Therefore, we performed RT-qPCR to measure the organ-specific expression level of Fam210a mRNA. Please see new Figure EV1A and updated Figure 6B. The organ enrichment of FAM210A expression is consistent between human and mouse heart and skeletal muscle (Figure 6B and EV1A). Interestingly, miR-574-5p is also highly expressed in the mouse heart, suggesting that FAM210A mRNA expression could be regulated at the transcriptional level prior to post-transcriptional regulation mediated by miR-574. Please see Page 16 in the text.*

Minor points:

1. In figure 2D and 3D, when showing the zoomed picrosirius red images, please pick a similar location from each heart.

Response: *As suggested by the reviewer, we picked a similar location from each heart to show the zoomed picrosirius red images. Please see updated Figure 2D and 3D.*

2. In figure 4, the overall effect of mimetic oligomers in rescuing cardiac function is not very robust. Please replace Echo images shown in figure 4D to better represent the result. Accordingly, please also moderate the statements related to the rescue potential of the used treatment.

Response: *As suggested by the reviewer, we replaced some images (more representative of the average value in the bar graphs) in updated Figure 4D to better represent the result and also modified the statements related to the rescue potential of the used treatment in the main text (TAC-induced cardiac hypertrophy and fibrosis were moderately reduced by miR-574-3p mimics; Cardiac function was partially or moderately recovered with miR-574-3p and miR-574-5p treatment at the end point; In addition, miR-574-5p/3p treatment moderately decreased ROS production...). In addition, we have also included LVIDd and LVIDs as well as cardiac output, stroke volume, and LV mass in the updated supplemental Tables.*

3. Also, apoptosis is not a major mechanism for cardiomyocyte cell death during pressure overload. It is likely that the few TUNEL positive nuclei observed in the myocardium might actually originate from non-myocyte populations. Perhaps including that assay as supplemental data would be better suited.

Response: *We agree with the reviewer that CM apoptosis is not a strong phenotype during pressure overload and TUNEL signal may not come from CM cells. As suggested by both reviewer 1 and 2, we have moved TUNEL experimental data to supplemental figures (Figure EV2E, EV3F), and modified the statement in the text (...cardiac apoptosis (though occurring at modest level under pressure overload)...).*

Referee #3 (Comments on Novelty/Model System for Author):

While the cardiac function of miR-574 is unknown and the authors make use of a new knockout

mouse model, the cellular function of miR-574 remains unclear and the in vivo relevance of the proposed mechanism is not investigated.

Since a global knockout is used and the mimics are also delivered to all cardiac cell types the authors would have to study the function of miR-574 in multiple cell types to study its true contribution to cardiac function and remodeling in vivo.

Response: *We highly appreciate this constructive and important suggestion from the reviewer. To address the reviewer's questions regarding the cellular function of miR-574, we measured miR-574-5p and miR-574-3p in isolated primary cardiomyocytes and cardiac fibroblasts from hearts under TAC surgery. We found that miR-574-5p and miR-574-3p were significantly induced in CMs and CFs, respectively and the other strand was only slightly induced in each cell type (new Figure EV1C). We have also performed in vitro experiments using the anti-miRNA inhibitors (loss-of-function) and miRNA mimics (gain-of-function) transfection in human AC16 CM cells and confirmed the anti-hypertrophic (Figure EV4C, D) and mitochondrial protective (Figure 8E, F) effects of miR-574-5p/3p. The anti-hypertrophic effect of miR-574-5p/3p mimics was also confirmed in primary mouse neonatal CMs (New Figure EV3G). More importantly, the anti-hypertrophic, anti-apoptotic, and mitochondrial protective activities of miR-574 were validated in isolated primary mouse adult CM cells from WT and miR-574 KO hearts (Figure EV2A, B and Figure 8G, H). In contrast, we did not observe obvious effects of both miR-574-5p and miR-574-3p on the proliferation of primary mouse CFs (new Figure EV3H). miR-574-3p (not miR-574-5p) reduced α -SMA protein expression in TGF- β -activated primary mouse CFs though both miRNA strands inhibited α -SMA protein expression at baseline (new Figure EV3I). These results suggest that exogenous miR-574-5p and miR-574-3p mimics may primarily target CMs as a synergistic pair and could also target CFs as a complementary mechanism. Please see Page 11 and 14 in the text. Vertical lines were drawn in the left margin area to highlight the major changes.*

To address the reviewer's question regarding the in vivo relevance of the proposed mechanism, we tested our hypothesis of miR-574-Fam210A axis influencing mitochondria encoded ETC protein expression and whether loss of miR-574 causes dysregulation of MEG and NEMG expression for the ETC complex. We measured the change of ETC proteins using IF in mice after 3-day post TAC (new Figure 8J and EV5A-C). We observed more significantly induced mitochondrial encoded ETC component protein expression as well as FAM210A in miR-574 KO hearts compared to WT hearts but the nuclear-encoded ETC component proteins were not significantly altered (new Figure 8J and EV5A-C). This imbalance between mitochondrial-encoded and nuclear-encoded protein expression at the early stage may cause mitochondrial stress and more severe cardiac pathological remodeling and heart dysfunction at the late stage, as we observed (Figure EV5D and Figure 3).

We have measured ROS production by DHE staining in heart frozen sections under ISO treatment, TAC surgery, and miR-574 mimics injections (Figure EV2C, EV2D, EV3D). We have also measured membrane potential by TMRE staining and ATP production in cultured primary mouse adult CM cells isolated from ISO-treated WT and miR-574 KO mice (Figure 8G, H and ATP production in the TAC plus miR-574 mimics treatment model (EV3E). We further measured the activity of complex I, II, III and IV in WT vs. miR-574 KO mice with Sham or TAC surgery after 2 months (New Figure EV5D) and found that the ETC complex activity is more severely affected in miR-574 KO vs. WT hearts, suggesting that unbalanced protein expression of ETC component proteins mentioned above could compromise their activity in the late stage. Please see Page 21 in the text.

Referee #3 (Remarks for Author):

In this manuscript the authors show that miR-574 is induced in the heart during cardiac disease in both mice and human. Global genetic deletion of miR-574 does not induce an overt

phenotype under baseline conditions but exacerbates remodeling during cardiac stress. Conversely, intravenous delivery of miR-574 mimic inhibits TAC induced pathological remodeling. The authors next define FAM210A as a direct target of both strands of miR-574 and show in vitro that this regulates cellular mitochondrial function.

MicroRNAs have been shown to be important players for cardiac biology and disease including mitochondrial function. While the observed in vivo effects are striking, some key data points are lacking to provide sufficient comfort regarding the proposed mechanism.

Some key issues:

- Which cardiac cells express miR-574 and how does this change during disease? Is this regulation different between mice and human?

Response: *We appreciate the constructive suggestions and questions from the reviewer. We address these questions separately as follows.*

Which cardiac cells express miR-574 and how does this change during disease?

We have added several experimental data to address the questions. New Figure EV1B shows that miR-574-5p is more dominantly expressed in cardiomyocytes (CMs) and also moderately expressed in cardiac fibroblasts (CFs), while miR-574-3p is moderately expressed in CMs and CFs. We isolated CMs and CFs in 3-day post TAC hearts from WT mice and measured the expression level of miR-574-5p and miR-574-3p. We found that miR-574-5p and miR-574-3p were significantly induced in CMs and CFs, respectively and the other strand was only slightly induced in each cell type (new Figure EV1C), suggesting miR-574-5p plays more dominant role in CMs during remodeling process. These observations are consistent with increased miR-574-5p and miR-574-3p observed in cultured adult mouse CM with ISO treatment (compared to vehicle treatment).

We further measured miR-574-5p and miR-574-3p expression level in the stressed heart at different time points post-TAC surgery, including 3-day, 7-day, 14-day and 21-day. We observed that miR-574-5p and miR-574-3p were induced by 2-3 folds on day 3 through day 21. In combination of results from Figure 1E (4 weeks after TAC surgery), we concluded that an increase in miR-574-5p and miR-574-3p occurs at both early and late stages of remodeling processes (d3, d7, d14, d21, and month 1). Please see Page 11 in the text.

Is this regulation different between mice and human?

We have shown that miR-574-5p and miR-574-3p are increased in human failing hearts similarly as in mouse failing hearts (Figure 1C-E). We don't have the capacity to culture live human heart derived primary CM and CF cells. It is clear that human FAM210A mRNA 3'UTR contains the target seed sequence sites of miR-574-5p and miR-574-3p (new Appendix Figure S2C). Please see Page 14 in the text. More importantly, we used human ventricular CM cell line AC16 to confirm the regulation of FAM210A via miR-574-5p and miR-574-3p. we have used anti-miR-574-5p, anti-miR-574-3p, and control anti-miR inhibitors to transfect AC16 human cardiomyocyte cells and measured the expression of mitochondrial encoded and nuclear encoded electron transport chain complex proteins and hypertrophic growth of AC16 cells under the treatment of ISO vs. vehicle (Figure 8A-D and 8I, Figure EV4A-D and EV4H, I). In vitro studies using human AC16 cells and in vivo studies using mouse genetic and disease models at least partially address the human relevance and evolutionary conservation of this regulatory mechanism. Therefore, we believe that the miR-574-5p/3p-mediated regulation of FAM210A is similar between mice and human.

- Which cardiac cell type is targeted by the miR-574 mimics and how does this effect the different cell types?

Response: Based on Figure 4B, both miR-574-5p and miR-574-3p reduced cardiac hypertrophy upon injection in vivo. We performed experiments to test if both miRNA strands could antagonize CM hypertrophy under ISO stimulation in vitro. New Figure EV3G showed that miR-574-5p and miR-574-3p could reduce myocyte hypertrophy in primary mouse neonatal cardiomyocyte cells. Moreover, Figure EV4C and EV4D showed that miR-574-5p and miR-574-3p could reduce myocyte hypertrophy in AC16 human cardiomyocyte cell line. These results suggest that miR-574-5p and miR-574-3p can target CM cells and reduce CM hypertrophy. Please see Page 14 and 18 in the text.

Figure 4C shows that injected miR-574-3p and miR-574-5p reduce cardiac fibrosis in vivo. We tested whether these anti-fibrotic effects were caused by autonomous or non-autonomous mechanism. We performed miR-574-5p and miR-574-3p treatment (compared to control miRNA) in primary adult CF cells to measure CF proliferation under baseline or upon TGF- β treatment following serum starvation (new Figure EV3H). We did not observe any significant effects of miR-574-5p and miR-574-3p on CF cell proliferation. We also measured α -SMA protein expression in TGF- β -treated CF cells, and found that miR-574-3p (not miR-574-5p) reduced α -SMA protein expression during TGF- β -triggered CF activation while both miRNA strands reduced α -SMA protein expression at baseline (new Figure EV3I). These observations suggest that the reduction of cardiac fibrosis might be partly caused by reduced CF activation and partly due to non-autonomous effects from normalized CM as a potential complementary mechanism. Please see Page 14 in the text.

We haven't tested other cardiac cell types such as immune cells since ISO and TAC mouse models are not considered as robust heart failure models driven by extensive immune cell infiltration and inflammation (like myocardial infarction). Endothelial cells or smooth muscle cells are not involved so much in ISO model even they could play a role in TAC model. We will further study the potential role of miR-574 in these immune and vascular cell types in the future study in vitro and in vivo.

- Which tissues besides the heart is the miR mimic delivered to and how does this influence the observed phenotype?

Response: Based on the delivery capacity of the nanoparticle-based in vivo transfection reagent

from Altogen Biosystems, miR-574 could be delivered to multiple organs in addition to heart. We measured miR-574-5p/3p level in kidney and liver from the mice and found that both miR-574-5p and miR-574-3p were also delivered to these two organs to some extent (Figures shown below). Figure EV3B suggests that no obvious toxicity or side-effects have been observed in kidney and liver. We cannot exclude the possibility that miR-574 delivery to the liver or kidney (or even other organs) may influence the observed cardioprotective phenotype. But based on our in vitro studies using CM and CF cells, we think that the effects of miR-574 in these two cardiac cell types could play a major role in the heart disease phenotypes, especially the autonomous anti-hypertrophic effects on CM cells and autonomous or potential non-autonomous anti-fibrotic effects on CF cells.

Figure legend: Expression levels of miR-574-3p and miR-574-5p in liver and kidney after injections of the miRNA mimics in WT mice for 2 and 4 days. All values are expressed as mean±SEM (n=4)

- The authors show that FAM210A is a direct target of both strands of miR-574. The authors are also able to show a regulation of FAM210A in the knockout mice. However, it remains unclear how this effects downstream regulators of mitochondrial function in vivo or how this effects mitochondrial function.

Response: *We performed transmission electric microscopy and found that mitochondria underwent more swelling, reduced cristae formation, and overall disarray in miR-574 KO mouse hearts than WT hearts upon the ISO stress (new Figure EV4F, G). This observation indicates that increased FAM210A in the miR-574 KO mice render the animals prone to more severe mitochondrial dysfunction under cardiac stresses.*

To test our hypothesis of miR-574-Fam210A axis influencing mitochondria encoded ETC protein expression and whether loss of miR-574 causes dysregulation of MEG and NEMG expression for the ETC complex, we measured the change of ETC proteins using IF in mice after 3-day post TAC (new Figure 8J and EV5A-C). We observed more significantly induced mitochondrial encoded ETC component protein expression as well as FAM210A in miR-574 KO hearts compared to WT hearts but the nuclear-encoded ETC component proteins were not significantly altered (new Figure 8J and EV5A-C). This imbalance between mitochondrial-encoded and nuclear-encoded protein expression in the early stage may cause mitochondrial stress and more severe cardiac pathological remodeling and heart dysfunction in the late stage as we observed (Figure EV5D and Figure 3).

We have measured ROS production by DHE staining in heart frozen sections under ISO treatment, TAC surgery, and miR-574 mimics injections (Figure EV2C, EV2D, EV3D). We have also measured membrane potential by TMRE staining and ATP production in cultured primary mouse adult CM cells isolated from ISO-treated WT and miR-574 KO mice (Figure 8G, H and ATP production in the TAC plus miR-574 mimics treatment model (EV3E). We further measured the activity of complex I, II, III and IV in WT vs. miR-574 KO mice with Sham or TAC surgery after 2 months (New Figure EV5D) and found that the ETC complex activity is more severely affected in miR-574 KO vs. WT hearts, suggesting that unbalanced protein expression of ETC component proteins mentioned above could compromise their activity in the late stage. Please see Page 21 in the text.

16th Oct 2020

Dear Prof. Yao,

Thank you for the submission of your revised manuscript to EMBO Molecular Medicine. We have now received the enclosed reports from the three referees who reviewed the new version of your manuscript. As you will see, they are supportive of publication, and I am thus pleased to inform you that we will be able to accept your manuscript pending the following final minor amendments:

1) Main manuscript text:

- Please answer/correct the changes suggested by our data editors in the main manuscript file (in track changes mode). This file will be sent to you in the next couple of days. Please use this file for any further modification.
- Please remove the vertical lines in the left margin area.
- Please move the "Material and methods" section after the "Discussion" and before the "Acknowledgements".
- Material and methods: Please include a statement that informed consent was obtained from all subjects and that the experiments conformed to the principles set out in the WMA Declaration of Helsinki and the Department of Health and Human Services Belmont Report. For the mice experiments, please indicate the housing and husbandry condition, as well as the age and gender of the mice. Please also indicate in the Material and Methods whether the cells were authenticated (when applicable) and tested for mycoplasma contamination.
- Please remove the Appendix figure legends from the main manuscript text (it should only be in the Appendix file). Please remove the legends from the figure files (they should be assembled together into one "Figure legends" section after the references).
- Please remove "data not shown". As per our guidelines on "Unpublished Data" the journal does not permit citation of "Data not shown". All data referred to in the paper should be displayed in the main or Expanded View figures. "Unpublished observations" may be referred to in exceptional cases, where these are data peripheral to the major message of the paper and are intended to form part of a future or separate study, the names of the persons that reported the observation should be listed in brackets. Personal communications (Author name(s), personal communications) must be authorised in writing by those involved, and the authorisation sent to the editorial office at time of submission.

2) Figures:

Each individual figure should fit into 1 page, including EV figures.

3) Source Data:

We encourage you to include the source data for figure panels that show essential data. Numerical data should be provided as individual .xls or .csv files (including a tab describing the data). For blots or microscopy, uncropped images should be submitted (using a zip archive if multiple images need to be supplied for one panel).

In particular, please provide Source Data for Fig. 6F.

4) Appendix:

We note that an consequent part of your material and methods has been placed in the Appendix.

We would suggest moving part of it back to the main manuscript text (such as the ISO injection, the TAC surgical model, the therapeutic model with miRNA mimics, etc...) to ensure better accessibility for the readers. (There will be no length restriction).

5) Checklist:

Please fill out part E/Human subjects, 11 and 12.

6) For more information: There is space at the end of each article to list relevant web links for further consultation by our readers. Could you identify some relevant ones and provide such information as well? Some examples are patient associations, relevant databases, OMIM/proteins/genes links, author's websites, etc...

7) Thank you for providing a synopsis. I slightly modified the text to fit our style and format, please let me know if you agree with the following:

The findings identify that miR-574 fine-tunes FAM210A expression and modulates mitochondrial encoded protein expression, thereby maintaining normal mitochondrial function and protecting the heart from cardiac stress induced pathological remodeling.

- miR-574-5p and miR-574-3p were induced in failing human and mouse hearts.
- miR-574 knockout mice showed severe pathological cardiac remodeling under stress, suggesting a cardioprotective role of that miR-574.
- Delivery of exogenous miR-574-5p/3p mimics antagonized pathological cardiac remodeling.
- FAM210A interacted with mitochondrial translation elongation factor EF-Tu and promoted mitochondrial encoded protein expression.
- miR-574 fine-tuned FAM210A expression and modulated mitochondrial encoded protein expression, thereby maintaining normal mitochondrial function.

The synopsis should be provided as a individual document.

Thank you for providing a nice synopsis image. Please make sure that the text remains readable when resized to 550px wide.

8) As part of the EMBO Publications transparent editorial process initiative (see our Editorial at <http://embomolmed.embopress.org/content/2/9/329>), EMBO Molecular Medicine will publish online a Review Process File (RPF) to accompany accepted manuscripts.

In the event of acceptance, this file will be published in conjunction with your paper and will include the anonymous referee reports, your point-by-point response and all pertinent correspondence relating to the manuscript. Let us know whether you agree with the publication of the RPF and as here, **IF YOU WANT TO REMOVE OR NOT ANY FIGURES** from it prior to publication.

I look forward to receiving your revised manuscript.

Yours sincerely,

Lise Roth

Lise Roth, PhD
Editor

To submit your manuscript , please follow this link:

Link Not Available

The system will prompt you to fill in your funding and payment information. This will allow Wiley to send you a quote for the article processing charge (APC) in case of acceptance. This quote takes into account any reduction or fee waivers that you may be eligible for. Authors do not need to pay any fees before their manuscript is accepted and transferred to our publisher.

***** Reviewer's comments *****

Referee #1 (Remarks for Author):

The authors have satisfactorily addressed all my previous concerns.

Referee #2 (Remarks for Author):

We thank the authors for being responsive to my comments and mitigating my concerns.

Referee #3 (Remarks for Author):

The authors were able to answer most of the issues raised and in doing so provided an improved version of the manuscript.

The authors performed the requested changes.

19th Nov 2020

Dear Prof. Yao,

We are pleased to inform you that your manuscript is accepted for publication in EMBO Molecular Medicine.

Before we transfer your manuscript to our publisher to be included in the next available issue of EMBO Molecular Medicine, could you please address the following:

- in your source data files, should figure 5D be 5F?
- in the source data for figure EV5, you mention that the order of the panels should be changed on the first slide of the powerpoint file, however I could not see the difference between the source data and the main manuscript.

Please let us know as soon as possible if the source data files have to be changed.

Congratulations on your interesting work!

Sincerely,

Lise Roth

Lise Roth, Ph.D
Scientific Editor
EMBO Molecular Medicine

*** ** IMPORTANT INFORMATION ** **

SPEED OF PUBLICATION

The journal aims for rapid publication of papers, using using the advance online publication "Early View" to expedite the process: A properly copy-edited and formatted version will be published as "Early View" after the proofs have been corrected. Please help the Editors and publisher avoid delays by providing e-mail address(es), telephone and fax numbers at which author(s) can be contacted.

Should you be planning a Press Release on your article, please get in contact with embomolmed@wiley.com as early as possible, in order to coordinate publication and release dates.

LICENSE AND PAYMENT:

All articles published in EMBO Molecular Medicine are fully open access: immediately and freely available to read, download and share.

EMBO Molecular Medicine charges an article processing charge (APC) to cover the publication costs. You, as the corresponding author for this manuscript, should have already received a quote with the article processing fee separately. Please let us know in case this quote has not been received.

Once your article is at Wiley for editorial production you will receive an email from Wiley's Author Services system, which will ask you to log in and will present you with the publication license form for completion. Within the same system the publication fee can be paid by credit card, an invoice, pro forma invoice or purchase order can be requested.

Payment of the publication charge and the signed Open Access Agreement form must be received before the article can be published online.

PROOFS

You will receive the proofs by e-mail approximately 2 weeks after all relevant files have been sent to our Production Office. Please return them within 48 hours and if there should be any problems, please contact the production office at embopressproduction@wiley.com.

Please inform us if there is likely to be any difficulty in reaching you at the above address at that time. Failure to meet our deadlines may result in a delay of publication.

All further communications concerning your paper proofs should quote reference number EMM-2020-12710-V3 and be directed to the production office at embopressproduction@wiley.com.

Thank you,

Lise Roth, Ph.D
Scientific Editor
EMBO Molecular Medicine

Corresponding Author Name: Peng Yao

Journal Submitted to: EMMO Molecular Medicine

Manuscript Number: EMM-2020-12710